EMBO
Molecular Medicine

# Synergistic effects of FGFR1 and PLK1 inhibitors target a metabolic liability in *KRAS*-mutant cancer

Zhang Yang[1,2,†] (iD), Shun-Qing Liang[1,†], Maria Saliakoura[3], Haitang Yang[1] (iD), Eric Vassella[4], Georgia Konstantinidou[3] (iD), Mario Tschan[4] (iD), Balazs Hegedüs[5], Liang Zhao[1], Yanyun Gao[1], Duo Xu[1], Haibin Deng[1], Thomas M Marti[1], Gregor J Kocher[1] (iD), Wenxiang Wang[6], Ralph A Schmid[1,*] & Ren-Wang Peng[1,**] (iD)

## Abstract

KRAS oncoprotein is commonly mutated in human cancer, but effective therapies specifically targeting KRAS-driven tumors remain elusive. Here, we show that combined treatment with fibroblast growth factor receptor 1 (FGFR1) and polo-like kinase 1 (PLK1) inhibitors evoke synergistic cytotoxicity in *KRAS*-mutant tumor models *in vitro* and *in vivo*. Pharmacological and genetic suppression of FGFR1 and PLK1 synergizes to enhance anti-proliferative effects and cell death in *KRAS*-mutant lung and pancreatic but not colon nor *KRAS* wild-type cancer cells. Mechanistically, co-targeting FGFR1 and PLK1 upregulates reactive oxygen species (ROS), leading to oxidative stress-activated c-Jun N-terminal kinase (JNK)/p38 pathway and E2F1-induced apoptosis. We further delineate that autophagy protects from PLK1/FGFR1 inhibitor cytotoxicity and that antagonizing the compensation mechanism by clinically approved chloroquine fully realizes the therapeutic potential of PLK1 and FGFR1 targeting therapy, producing potent and durable responses in *KRAS*-mutant patient-derived xenografts and a genetically engineered mouse model of *Kras*-induced lung adenocarcinoma. These results suggest a previously unappreciated role for FGFR1 and PLK1 in the surveillance of metabolic stress and demonstrate a synergistic drug combination for treating *KRAS*-mutant cancer.

**Keywords** autophagy; fibroblast growth factor receptor 1; *KRAS*-mutant cancer; polo-like kinase 1; synthetic lethal vulnerability
**Subject Category** Cancer

## Introduction

More than 30% of non-small cell lung cancer (NSCLC) patients have *KRAS* mutations (Prior *et al*, 2012; Barlesi *et al*, 2016). Unlike NSCLC driven by other less frequent oncogenic drivers (e.g., *EGFR*, *ALK*, *MET1*, and *ROS1*) that significantly benefit from selective kinase inhibitors (Reck & Rabe, 2017), there are still no clinically approved therapies that specifically target *KRAS*-mutant cancers (Cox *et al*, 2014). Although immune checkpoint inhibitors of programmed death 1 (PD1) and programmed death ligand 1 (PD-L1) have been approved for NSCLC, they fail to discriminate *KRAS*-mutant from other NSCLC (Jeanson *et al*, 2019). Covalent KRAS inhibitors have demonstrated promise in preclinical models, but they are only effective against a specific *KRAS*-mutant allele and additional agents are needed to optimize the anti-cancer efficacy (Ostrem *et al*, 2013; Janes *et al*, 2018; Molina-Arcas *et al*, 2019). Targeting KRAS downstream effectors, e.g., the mitogen-activated protein kinase (MAPK) RAF/MEK/ERK and phosphatidylinositol 3-kinase (PI3K)/AKT/mTOR, has been widely pursued, but the pleiotropic nature and complex interconnection of individual signaling cascade and toxicity ensuing from sustained inhibition of multiple pathways has hindered the translational potential of the strategy (Shimizu *et al*, 2012; Samatar & Poulikakos, 2014; Manchado *et al*, 2016). Consequently, identification of new therapeutic targets for innovative treatment strategies tailored to *KRAS*-mutant cancers still represents a pressing need (Yang *et al*, 2019a).

One strategy for targeting *KRAS*-driven tumors is to exploit collateral damage contextually induced by a mutant KRAS, in light of the concept that mutant KRAS alters physiological biochemical networks and induces cellular stresses (genotoxic, proteotoxic, and metabolic), rendering *KRAS*-mutant cancer cells particularly reliant

---

1 Division of General Thoracic Surgery, Department for BioMedical Research (DBMR), Inselspital, Bern University Hospital, University of Bern, Bern, Switzerland
2 Department of Thoracic Surgery, Fujian Medical University Union Hospital, Fuzhou, China
3 Institute of Pharmacology, University of Bern, Bern, Switzerland
4 Institute of Pathology, University of Bern, Bern, Switzerland
5 Department of Thoracic Surgery, University Medicine Essen - Ruhrlandklinik, University Duisburg-Essen, Essen, Germany
6 The Second Thoracic Surgery Department, Hunan Cancer Hospital and The Affiliated Cancer Hospital of Xiangya School of Medicine, Central South University, Changsha, China
*Corresponding author. Tel: +41 0 31 632 37 45; E-mail: ralph.schmid@insel.ch
**Corresponding author. Tel: +41 0 31 632 40 81; E-mail: renwang.peng@insel.ch
†These authors contributed equally to this study

on stress-remedy signaling for survival (Luo *et al*, 2009; Downward, 2015; Nagel *et al*, 2016). Oncogenic KRAS activation has been shown to upregulate mitochondrial reactive oxygen species (ROS), and this increase in ROS is essential for *KRAS*-induced tumorigenicity (Solimini *et al*, 2007; Weinberg *et al*, 2010; De Raedt *et al*, 2011). However, persistent ROS accumulation is detrimental, as it can overwhelm the cellular anti-oxidant response and evoke cell death. As a result, perturbations of stress-adaptive mechanisms that enable cancer cells to grow and survive under adverse conditions constitute a promising strategy to target *KRAS*-mutant cancer (Genovese *et al*, 2017; Yang *et al*, 2019b).

PLK1 is a serine/threonine kinase orchestrating diverse events throughout mitotic progression: from G2/M transition and mitotic entry to spindle formation, centrosome maturation, chromosome segregation, and cytokinesis (Zitouni *et al*, 2014). PLK1 has also been implicated in non-mitotic functions by regulating ataxia-telangiectasia mutated (ATM) /CHK2 and ATM- and Rad3-Related (ATR)/CHK1 checkpoint activity in response to stress stimuli such as DNA damage (Smits *et al*, 2000; Takaki *et al*, 2008; van Vugt *et al*, 2010; Yata *et al*, 2012; Li *et al*, 2017). PLK1 is highly expressed in malignant tumors but scarcely detectable in normal tissues, and its expression correlates with poor patient survival (Strebhardt, 2010; Medema *et al*, 2011). Importantly, PLK1 is a synthetic lethal target in cancer driven by mutant *KRAS* (Luo *et al*, 2009), although how PLK1 crosstalks with oncogenic KRAS signaling remains incompletely understood. All these findings have aroused enormous interest for targeting PLK1 in cancer and prompted the development of selective PLK1 inhibitors (Rudolph *et al*, 2009; Sebastian *et al*, 2010), including volasertib (BI6727) with a breakthrough status for leukemia and a favorable safety profile (Döhner *et al*, 2014). However, the utility of PLK1-targeted therapy alone is rather limited (Gutteridge *et al*, 2016), highlighting the need to identify complementary targets for the development of combination strategies.

Synergistic combination therapies have been widely pursued due to their potential to augment effectiveness and selectivity, to decrease individual drug dosage, to reduce the development of drug resistance, and possibly to avoid toxicity (Lehár *et al*, 2009). Here, we reported, for the first time, that combined treatment with FGFR1 and PLK1 inhibitors evokes synergistic cytotoxic activity in *KRAS*-mutant lung and pancreatic cancer models *in vitro* and *in vivo*. The drug synergy is ascribed to a metabolic liability for increased ROS that comes along with oncogenic KRAS. We further showed that autophagy limits FGFR1/PLK1 inhibitor efficacy and that addition of chloroquine leads to greater efficacy. These results point to a previously unappreciated role for FGFR1 and PLK1 in coping with oxidative stress and demonstrate a new therapeutic rationale for *KRAS*-mutant cancers.

## Results

### Drug synergy between FGFR and PLK1 inhibitors is conditionally induced by oncogenic KRAS activation

Despite the indispensability of PLK1 in mutant *KRAS*-driven cancer (Luo *et al*, 2009), efficacy of PLK1 inhibitors as monotherapy has been thwarted by poor response rates of patients (Sebastian *et al*,

2010; Gutteridge *et al*, 2016). Hypothesizing that sustained PLK1 inhibition is essential but additional targets are needed for effective targeting of *KRAS*-mutant cancers, we undertook a chemical synthetic lethal screen for small-molecule compounds that were able to enhance anti-proliferative effects of BI2536, a selective and clinically advanced PLK1 inhibitor (Sebastian *et al*, 2010). We chose a pool of compounds ($n = 21$), including FDA-approved drugs, clinical candidates, and previous screening targets that individually interrogate important targets and/or cellular processes in cancer (Appendix Table S1).

To minimize the potential effects of other alterations co-occurring with mutant KRAS, we performed the screen in isogenic bronchial epithelial BEAS-2B cells that either express a mutant $KRAS^{G12V}$ allele, designated as BEAS-2B-KRAS, or harbor wild-type *KRAS* (referred to as BEAS-2B) that were generated in our previous study (Langsch *et al*, 2016). BEAS-2B-KRAS cells showed elevated p-ERK1/2 and p-AKT, increased cell proliferation, and a greater dependency on PLK1 (Appendix Fig S1A and B), indicating that expression of the mutant KRAS is sufficient to activate KRAS downstream signaling and, more importantly, to induce KRAS-dependent oncogenic phenotypes.

We probed the sensitivity profile of BEAS-2B-KRAS and BEAS-2B cells to individual compounds, alone and in combination with BI2536, across a multiple-point concentration range that had been optimized based on the half maximal inhibitory concentration ($IC_{50}$) of each compound in BEAS-2B-KRAS and BEAS-2B cells (Appendix Tables S1 and S2). The effectiveness of all drug combinations ($n = 21$) was measured by combination index (CI) and fraction affected (Fa), whereby synergistic (CI < 1), additive (C = 1), or antagonistic (CI > 1) of each drug pair could be calculated (Fig 1A and B; Appendix Fig S1C).

Several compounds, i.e., the FGFR inhibitor AZD4547 (Zhang *et al*, 2012), fasudil (ROCK inhibitor), ponatinib (multi-tyrosine kinase inhibitor), QNZ (NF-KB inhibitor), and onalespib (HSP90 inhibitor), synergistically enhanced the anti-proliferative effect of BI2536 in BEAS-2B-KRAS cells, with the most potent synergy conferred by AZD4547 (CI = 0.25), followed by QNZ (CI = 0.31) and ponatinib (CI = 0.38) (Fig 1B; Appendix Fig S1C). However, ponatinib, QNZ and onalespib also strongly enhanced BI2536 effects in BEAS-2B cells, with the CI value equal to 0.81, 0.78, and 0.65, respectively (Fig 1B; Appendix Fig S1C), excluding their potential as therapeutic combinations. In sharp contrast, AZD4547 in combination with BI2536 displayed a superior combinatorial index (CI = 0.25; Fa = 0.97) in BEAS-2B-KRAS cells, but strikingly discriminated the isogenic BEAS-2B counterpart (CI = 1.12; Fa = 0.51), indicating the potency and high selectivity of AZD4547/ BI2536 synergy in BEAS-2B-KRAS cells only (Fig 1B; Appendix Fig S1C). Interestingly, we observed a strong synergy between BI2536 and fasudil in BEAS-2B-KRAS cells (CI = 0.57; Fa = 0.88), although the two drugs also synergized (CI = 0.93; Fa = 0.51) in inhibiting BEAS-2B cell proliferation (Fig 1B; Appendix Fig S1C), suggesting that the combinatorial efficacy of fasudil/BI2536 is not restricted to *KRAS*-mutant setting and may come along with toxicity for normal tissue, which is in line with a previous report that fasudil potentiates PLK1 inhibition in *KRAS*-mutant cancer (Wang *et al*, 2016).

The activity and selectivity of the synergy between FGFR1 and PLK1 inhibitors were equally pronounced when other agents, i.e.,

BGJ398 (Nogova *et al*, 2017) and BI6727/volasertib (Döhner *et al*, 2014) against FGFR and PLK1, respectively, were used (Fig 1C). Alternative approaches, i.e., long-term clonogenic assay, showed similar results (Fig 1D). Importantly, Annexin V/propidium iodide (PI)-based apoptosis assay revealed that combined AZD4547 and BI2536 synergistically enhanced apoptosis in BEAS-2B-KRAS but not in BEAS-2B cells (Fig 1E). Taken together, these results demonstrate a synergistic interaction between FGFR and PLK1 inhibitors that is conditionally induced by oncogenic KRAS activation.

## Genetic and pharmacological inhibition of FGFR1 and PLK1 synergizes to enhance anti-proliferative effects and drive apoptosis in *KRAS*-mutant lung and pancreatic cancer cells

Next, we evaluated the therapeutic potential of co-targeting PLK1 and FGFR in *KRAS*-mutant cancer cells, with *KRAS* wild-type cancer cells, malignant pleural mesothelioma (MPM) cells, and non-transformed normal cells (Appendix Table S3) tested in parallel for comparison. AZD4547 profoundly and synergistically enhanced the

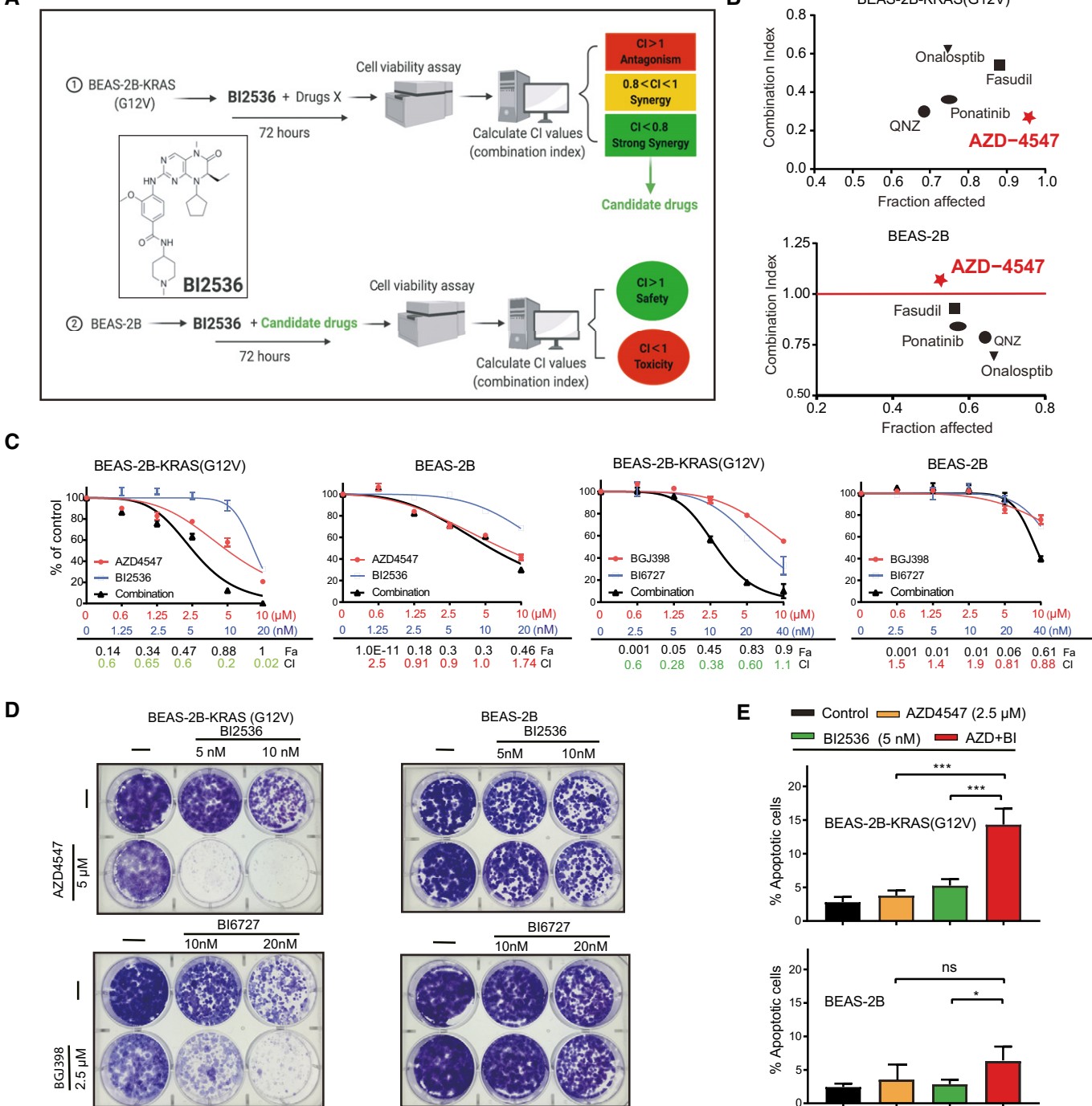

**Figure 1.**

**Figure 1. A synergistic drug combination conditionally activated by mutant KRAS.**

A   Schematic of chemical synthetic vulnerability screens. Clinical related drugs ($n = 21$) were combined with BI2536 to treat BEAS-2B-KRAS cells harboring a $KRAS^{G12V}$ allele. Cell viability was determined 72 h post-treatment, with combination index (CI) calculated by CalcuSyn software: CI > 1, antagonism; 1 > CI > 0.8, synergy; and CI < 0.8, strong synergy. Drug combinations leading to strong synergy (CI < 0.8) in BEAS-2B-KRAS were further tested in KRAS wild-type BEAS-2B cells, where CI < 1 indicates toxicity and CI > 1 safety.

B   The Fa-CI plot of drug combinations (candidate drug plus BI2536) in KRAS-mutant (BEAS-2B-KRAS) and wild-type (BEAS-2B) cells. The data are mean values of two independent experiments ($n = 2$) where all drugs are used at their $IC_{50}$ determined in BEAS-2B-KRAS cells. Note that combination treatment with AZD4547 and BI2536 exhibits the leading therapeutic efficacy and safety profile, manifested by the greatest synergy (CI = 0.25) in BEAS-2B-KRAS, while the two drugs are antagonistic in BEAS-2B cells (CI > 1).

C   Dose–response curves of BEAS-2B and BEAS-2B-KRAS cells treated for 72 h with inhibitors of FGFR (AZD4547, BGJ398) and PLK1 (BI2536, BI6727), alone or in combination. Values of Fa and CI are shown underneath, with CI < 1.0 indicating synergistic effect. Data were from three independent experiments ($n = 3$); error bar: SD.

D   BEAS-2B and BEAS-2B-KRAS treated with the indicated drugs, alone or in combination for 72 h, were cultured in drug-free medium for additional 7–21 days. Surviving cells after the treatment were fixed and visualized by crystal violet staining.

E   Apoptotic assay of BEAS-2B and BEAS-2B-KRAS cells treated with vehicle (DMSO), AZD4547 (2.5 μM), and BI2536 (5 nM), alone or in combination for 72 h. Data are presented as mean ± SD ($n = 3$). *$P < 0.05$ and ***$P < 0.001$ by two-way ANOVA with Tukey's multiple comparison test.

anti-proliferative effect of BI2536 in KRAS-mutant lung and pancreatic cancer cells, although AZD4547 (5 μM) and BI2536 (5 nM) alone at the tested dosage only mildly to moderately affected cell growth (Fig 2A and B). Notably, the same effect was not observed in KRAS-mutant colon cancer cells, nor in KRAS wild-type lung cancer cells, MPM cells, or normal human epithelial cells (Fig 2A and B). Western blot showed that AZD4547 (5 μM) and BI2536 (2.5 nM), alone and in combination, effectively inhibited FGFR and PLK1 downstream effectors, e.g., p-FRS2 (T436), p-AKT (S473), and p-PLK1 (T210) in A549 and H358 cells (Appendix Fig S1D–F), confirming on-site effects of the inhibitors.

Interrogating the CI values against predominant oncogenic drivers and cancer lineages indicated a remarkable association between FGFR1/PLK1 inhibitor synergy (CI < 1) with mutant KRAS, but not with mutations in EGFR, NRAS, BRAF, MET, or ALK, and a restriction of the synergy to KRAS-mutant lung and pancreatic cancer cells but not to KRAS-altered colon cancer cells, as indicated by the CI values (Fig 2B) and the fold change (increase) of sensitivity to BI2536 in the presence of AZD4547 (Appendix Figure S2A), although KRAS-mutant cancer cells, including those of colon origin, were generally more sensitive to PLK1 inhibition (BI2536) than KRAS-WT cancer cells (Appendix Figure S2B), consistent with the essential role of PLK1 in KRAS-mutant cancer (Luo et al, 2009). Notably, the synergy occurred across a wide dose range of AZD4547

and BI2536 in KRAS-mutant lung and pancreatic (H358, H441, A549, and MIAPaCa-2), but not in KRAS-mutant colon (SW620, DLD-1) nor KRAS-WT lung (EBC-1, H1993) cancer cells (Appendix Fig S2C).

The combinatorial effect of AZD4547/BI2536 in KRAS-mutant lung and pancreatic cancer cells was confirmed by clonogenic and apoptotic assay, where the drug combination markedly enhanced growth inhibition and led to a significantly greater percentage of apoptotic cells than single agents (Fig 2C and D), which did not occur in SW620, DLD-1, or EBC-1 cells (Appendix Fig S2D).

Moreover, while AZD4547 (5 μM) and BI2536 (5 nM) alone induced cell-cycle arrest at the G1 (55.8% by vehicle versus 80.9% by AZD4547) and G2/M (29.56% by vehicle versus 55.44% by BI2536), respectively, combined treatment with AZD4547/BI2536 rendered an even greater majority of the cells (81.43%) arrested at the G2/M (Appendix Fig S2E).

We then turned toward genetic evidence. RNA interference (RNAi)-mediated depletion of PLK1 and FGFR1 (Appendix Fig S2F and G) significantly enhanced the anti-proliferative effect elicited by single knockdown of PLK1 or FGFR1 in lung (A549, H358, H441) and pancreatic (MIA PaCa-2) cells, but not in the colon (SW620) cancer cells (Fig 2E), in line with the results of pharmacological inhibition (Fig 2A–D). Notably, downregulation of FGFR2 or FGFR3, which alone moderately inhibited A549 proliferation, failed to

**Figure 2. Combined inhibition of FGFR1 and PLK1 induces synergistic cytotoxicity in KRAS-mutant lung cancer and pancreatic cells.**

A, B   KRAS-mutant ($n = 17$) and wild-type ($n = 14$) cancer cell lines treated for 72 h with AZD4547 (5 μM) and BI2536 (5 nM), alone or in combination. The percentage of viable cells was color coded in a heatmap (A), with the combination index (CI) of AZD4547/BI2536 across the cell lines with different driver mutations shown in (B). CI < 0.8 indicates strong synergistic effect.

C   Clonogenic assay of KRAS-mutant lung and pancreatic cells (MIA PaCa-2, H2122, H2009, H23, H358, and A549) treated with control (DMSO), AZD4547 (5 μM), and BI2536 (5 nM), alone or in combination.

D   Apoptotic assay of KRAS-mutant cancer cells (H358, H441, A549, and SW620) and KRAS-WT lung cancer cells (EBC-1) treated with vehicle (DMSO), AZD4547 (5 μM), and BI2536 (5 nM), alone or in combination for 48h. Data are presented as mean ± SD ($n = 3$). ***$P < 0.001$ and ****$P < 0.0001$ by two-way ANOVA with Tukey's multiple comparison test.

E   KRAS-mutant lung (A549, H358, H441), pancreatic (MIAPaCa-2), and colon (SW620) cancer cells were transfected with control siRNAs and FGFR1- and PLK1-specific siRNAs, alone or in combination. Cell viability was determined 72 h post-transfection. Data are presented as mean ± SD ($n = 3$). **$P < 0.01$, ****$P < 0.0001$, and ns $P > 0.05$ by two-way ANOVA with Tukey's multiple comparison test. ns, no significance.

F   Growth curve of a KRAS-mutant patient-derived xenograft (BE564T) treated with vehicle, AZD4547 (10 mg/kg/per day), and BI6727 (5 mg/kg/per day), alone or in combination. *$P < 0.05$ by one-way ANOVA with Tukey's multiple comparison test. Data are the mean of tumor volume of each group (5 mice/group); error bar: SD.

G, H   Relative tumor volume (G) and weights (H) of the PDX (BE564T) after the treatment. *$P < 0.05$ by unpaired two-tailed t-test. (H) Data are the mean of tumor weight of each group (5 mice/group); error bar: SD.

Source data are available online for this figure.

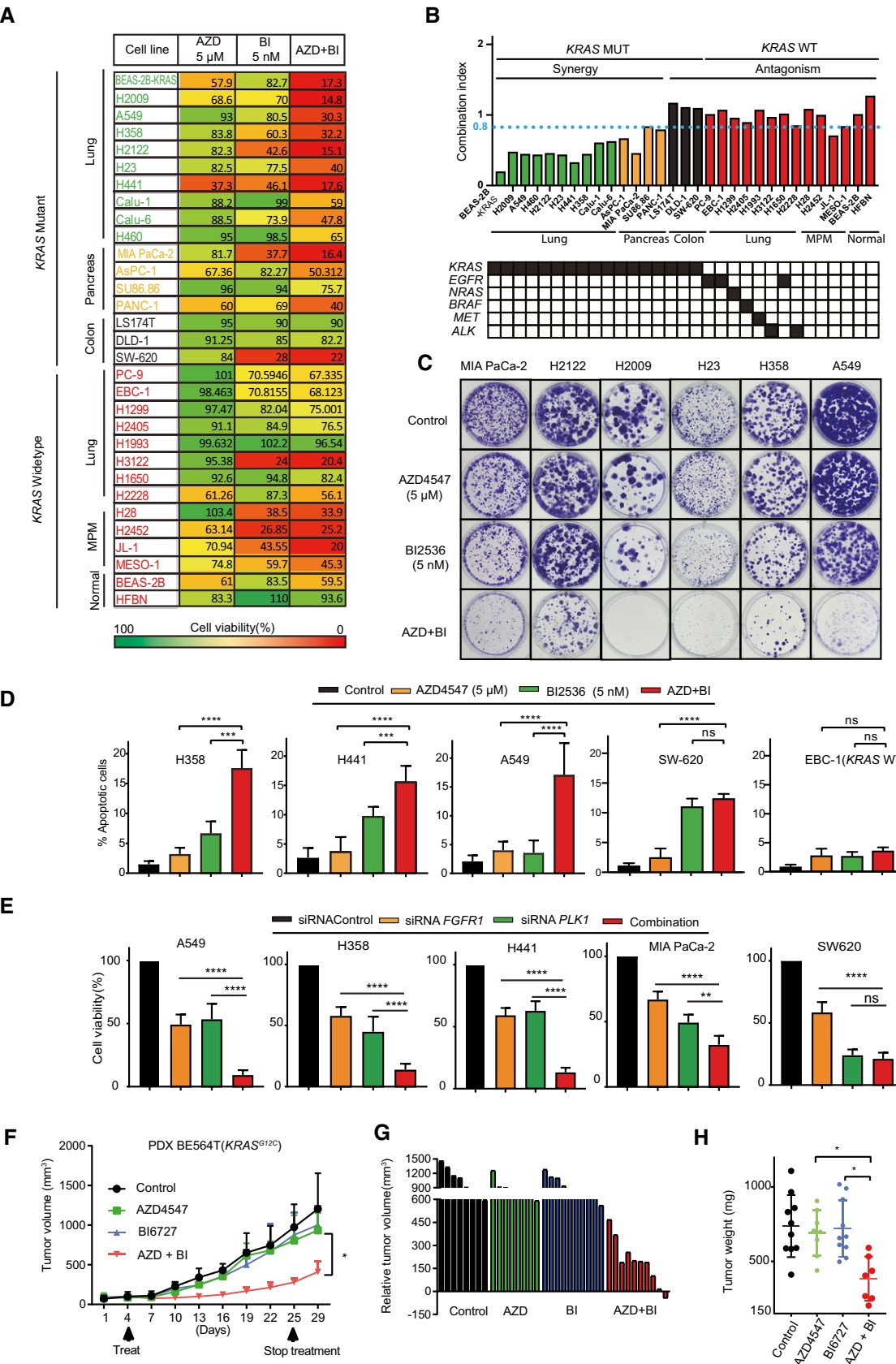

**Figure 2.**

significantly enhance the effect of *PLK1* knockdown (Appendix Fig S2H and I), which may be because A549 mainly expressed FGFR1 (Appendix Fig S2J). Whether differential expression of FGFR1/2/3 or the effector FRS2 (Appendix Fig S2J) correlated with sensitivity of cancer cells to FGFR inhibitors remains to be investigated. Nevertheless, the genetic evidence is in line with our pharmacological results and confirms the synergy of PLK1 and FGFR inhibition in *KRAS*-mutant lung and pancreatic cancer cells.

Given the biological implications of combined FGFR1/PLK1 inhibition, we then validated our *in vitro* results in a *KRAS*-mutant patient-derived xenograft (PDX). Compared to AZD4547 and BI6727 alone that had only minor anti-tumor effects, the drug combination showed significantly ($P < 0.05$) greater inhibition of tumor growth (Fig 2F–H), paralleled by significantly suppressed proliferation (Ki-67) and increased apoptosis (caspase-3), despite similar changes in mice body weights compared to single agents after 3-week treatment (Appendix Fig S2K–M). Thus, combined FGFR1/PLK1 inhibition evokes synergistic cytotoxicity, which selectively impairs proliferation, enhances apoptosis, thwarts cell-cycle progression *in vitro,* and potently suppresses *KRAS*-mutant lung tumor growth *in vivo*.

### Combined FGFR/PLK1 inhibition abolishes ROS homeostasis in *KRAS*-mutant cancer cells

In light of the potent *in vitro* and *in vivo* treatment response, next we explored the molecular basis underlying the synergistic effect of FGFR1/PLK1 inhibition. We noticed that genetic depletion of FGFR1 increased PLK1 activity and PLK1 reduction upregulated FGFR1 in *KRAS*-mutant lung and pancreatic cancer cells (Appendix Fig S2F and G), suggesting that FGFR1- and PLK1-mediated pathways reciprocate in *KRAS*-mutant lung cancer. Supporting this finding, correlation analysis of gene expression data of a TCGA cohort of *KRAS*-mutant lung cancer patients ($n = 141$) revealed that the mRNA level of *PLK1* is significantly negatively correlated with that of FGFR pathway genes, in particular strongly correlated (Spearman's $\rho > 0.3$) with *FGFR2*, *FGFR3,* and several *FGFs* (e.g., *FGF14, FGF22*), the cognate ligands of FGFR and triggers of FGFR-mediated signaling (Appendix Fig S3A). In *KRAS*-mutant pancreatic cancer cohort ($n = 133$), a strong correlation between *PLK1* and *FGF9, FGF14* was observed, and in *KRAS*-mutant colon cancer ($n = 170$), only *FGF14* strongly correlated with *PLK1* (Appendix Fig S3B and C). We also observed a strong negative correlation between *PLK1* and *FGFR3* in *EGFR*-mutant lung cancer cohort (Appendix Fig S3D), whereas there are no such correlations in *BRAF*-mutant lung cancer (Appendix Fig S3E).

To systematically probe the cellular processes that might represent the target of FGFR1/PLK1 inhibition, we performed gene set enrichment analysis (GSEA) based on a published study (Nakanishi *et al*, 2015), whereby *FGFR1*-amplified H520 and H1581 cells treated with the FGFR inhibitor CH5183284 were transcriptomically analyzed. Given that KRAS is the key downstream effector of FGFR1, we assumed that cancer cells harboring oncogenic activation of KRAS or FGFR1 have common deregulated cellular processes, which is supported by our recent study demonstrating a synergy between FGFR1 and PLK1 inhibitors in *FGFR1*-amplified lung cancer cells (Zhang *et al*, 2012). CH5183284 significantly suppressed mTORC1 signature in H520 and H1581 cells (Appendix Fig S4A), which held true in A549 and H358 cells, as treatment with AZD4547 or *FGFR1*-specific siRNAs inhibited mTORC1 activity (e.g., p-AKT, p-mTOR, and p-S6) (Appendix Fig S4B and C). CH5183284 also significantly blunted G2/M checkpoint, MYC, and E2F1 gene signatures (Appendix Fig S4A), suggesting a possible role for FGFR1 in MYC- and E2F1-mediated transcriptional programs.

Notably, FGFR1 blockage significantly upregulates peroxisome- and lysosome-related genes that are implicated in redox and autophagy, respectively (Fig 3A), suggesting a regulatory role for FGFR1 in these processes. Oncogenic KRAS is known to induce metabolic changes and alters cellular signaling that both can increase the production of intracellular ROS (Weinberg *et al*, 2010), and we thus investigated the possibility of oxidative damage upon FGFR1/PLK1 inhibition. While the basal level of intracellular ROS was markedly higher in BEAS-2B-KRAS than in BEAS-2B cells, consistent with the previous finding (Weinberg *et al*, 2010), combined treatment with AZD4547 and BI2536 significantly upregulated ROS levels in BEAS-2B-KRAS but not in BEAS-2B cells (Appendix Fig S4D and E). Importantly, there was significant increase of intracellular ROS in lung (H358, A549) and pancreatic (MIA PaCa-2) cancer cells following the treatment with AZD4547/BI2536 combination (Fig 3B). In contrast, BI2536 alone was sufficient to induce high levels of ROS in *KRAS*-mutant colon cancer cells (SW620) and addition of AZD4547 failed to further increase BI2536-induced ROS in the cells (Appendix Fig S4F), whereas in *KRAS*-WT lung cancer cells (EBC-1),

**Figure 3. Combined treatment with FGFR1/PLK1 inhibitors upregulates intracellular ROS.**

A   FGFR inhibition upregulates peroxisome and lysosome signatures. Gene set enrichment analysis (GSEA) is based on the GEO dataset GSE73024, whereby H1581 and H520 cells treated with CH5183284/Debio 1347, a selective FGFR inhibitor.

B   Flow cytometry-based measure of ROS in H358, A549, and MIA PaCa-2 cells treated for 24 h with AZD4547 (5 μM) and BI2536 (5 nM) in the presence or absence of NAC (5mM). Quantification of relative ROS levels was shown in the right. Data are presented as the mean ± SD ($n = 3$). ****$P < 0.0001$ by two-way ANOVA with Tukey's multiple comparison test.

C   Apoptotic assay of H358 and A549 cells treated for 48 h with vehicle (DMSO), AZD 4547 (5 μM), and BI2536 (5 nM), in the presence or absence of NAC (5 mM). Data are presented as mean ± SD ($n = 3$). *$P < 0.05$, **$P < 0.01$, ***$P < 0.001$, and ns $P > 0.05$ by two-way ANOVA with Tukey's multiple comparison test.

D   Clonogenic assay of H358, A549, and MIA PaCa-2 cells treated with AZD4547, BI2536, and NAC, alone or in combination as indicated.

E   Immunoblots of H358 and A549 cells treated for 24 h with vehicle (DMSO), AZD4547 (5 μM), and BI2536 (5 nM), in the presence or absence of NAC (5 mM).

F   Immunoblots of H358 cells treated for 24 h with vehicle (DMSO), AZD4547 (5 μM), and BI2536 (5 nM), in the presence or absence of Q-VD-OPh (20 μM).

G, H   Apoptotic (G) and clonogenic (H) assay of H358 cells treated for 48 h with vehicle, AZD4547 (5 μM), and BI2536 (5 nM), in the presence or absence of Q-VD-OPh (20 μM). Data shown as mean ± SD ($n = 3$). ***$P < 0.001$ and ****$P < 0.0001$ by two-way ANOVA with Tukey's multiple comparison test.

Source data are available online for this figure.

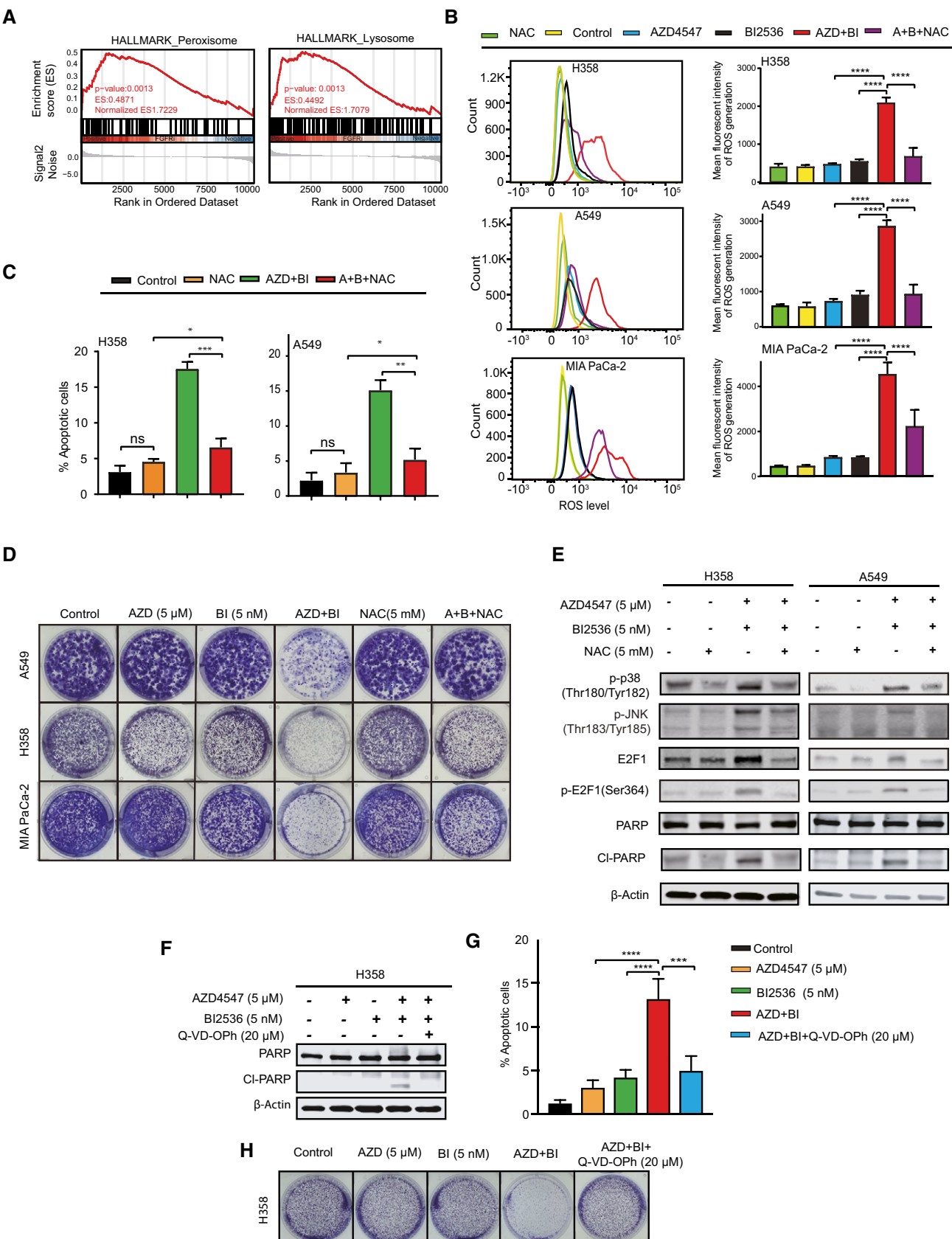

**Figure 3.**

the combination only slightly elevated ROS compared to single agents (Appendix Fig S4F).

Next, we determine whether ROS is the cause of cell death rather than an indirect effect of FGFR1/PLK1-induced cytotoxicity. The anti-oxidant N-acetyl cysteine (NAC), a ROS scavenger that had little to no toxicity at the selected drug dose (Fig 3C), effectively reduced the ROS level in AZD4547/BI2536-treated H358, A549, and MIA PaCa-2 cells (Fig 3B). Notably, NAC was able to neutralize the cytotoxicity of AZD4547/BI2536 in H358 and A549 cells (Fig 3C and D) and to decrease the cleavage of poly (ADP-ribose) polymerase (PARP), an apoptotic marker, under the same condition (Fig 3E). Supporting a critical role for apoptosis in the cytotoxicity of AZD4547/BI2536, addition of a pan-apoptotic inhibitor (Q-VD-OPH) to the combination decreased Cl-PARP expression, apoptotic cell death, and anti-proliferative effects in H358 cells (Fig 3F–H). Thus, FGFR1/PLK1 inhibitor-induced ROS is important for drug synergy in *KRAS*-mutant cells.

## ROS-activated JNK/p38/E2F1 axis contributes to synergistic activity of FGFR/PLK1 inhibitors

The stress-activated kinases c-Jun N-terminal kinase (JNK) and the mitogen-activated protein kinase (MAPK) p38 play a key role in response to stress insults (Arimoto et al, 2008; Wu et al, 2020), and we therefore explored whether elevated ROS upon FGFR/PLK1 inhibition can activate JNK and p38 in *KRAS*-mutant cancer cells.

We detected significant increase in the phosphorylation of p38 (p-p38), JNK (p-JNK), and E2F1 (p-E2F1), a transcription factor involved in stress-activated apoptosis, as well as cleavage of PARP and caspase-3 (CI-PARP, CI-caspase-3), but decrease in the anti-apoptotic BCL-2 in H358 and A549 cells treated for 72 h with AZD4547 (5 μM) and BI2536 (5 nM) drug combination (Figs 4A and 3E). Notably, combined AZD4547/BI2536 upregulated p-JNK, p-p38, p-E2F1, and CI-PARP in a mutant KRAS-dependent fashion, similar to ROS (Appendix Fig S4D and E), as AZD4547/BI2536-induced increase of p-p38, p-JNK, p-E2F1, and Cl-PARP in BEAS-2B-KRAS cells was not observed in BEAS-2B cells (Appendix Fig S4G). Similar results were observed in PDX tumors, whereby the combination treatment not only inhibited FGFR1 signaling (reduced p-FRS2), but also substantially increased p-JNK, γH2AX, and p-E2F1 and concomitantly decreased anti-apoptotic BCL-2 compared to single agents (Appendix Fig S4H).

Notably, the drug combination (AZD4547/BI2536) showed very similar effects on the expression of p38, JNK, E2F1, and Cl-PARP in SW620 cells as BI2536 alone, although AZD4547 (5 μM) and

BI2536 (5 nM) inhibited their respective targets (e.g., p-FRS2, p-AKT, p-PLK1) (Appendix Fig S4I). Moreover, the combination failed to activate the p38/JNK-E2F1 axis and induce apoptosis, but successfully evoked autophagy (LC3-II expression) in EBC-1 cells (Appendix Fig S4I), in contrast to the scenario observed in *KRAS*-mutant lung cancer cells. Importantly, ROS blockade with NAC markedly reduced p-p38, p-JNK, p-E2F1, and CI-PARP in AZD4547/BI2536-treated (72 h) H358 and A549 cells (Fig 3E), and addition of NAC substantially rescued AZD4547/BI2536/HCQ toxicity in A549 cell, and in H358 cells as well, despite to a lesser extent (Appendix Fig S4J).

We went further to assess the role of E2F1 in AZD4547/BI2536-evoked synergistic effects, given that E2F1 activity (p-E2F1) is upregulated by FGFR/PLK1 inhibition (Figs 3E and 4A). While SP600125 and SB203580, selective inhibitors of JNK and p38, respectively, only marginally altered E2F1 or cleaved PARP and had no or mild effects on H358 cell proliferation (Appendix Fig S4K–N), the concomitant presence of SP600125 and SB203580 strikingly suppressed total E2F1, p-E2F1, and CI-PARP (Fig 4B) and significantly abrogated the anti-proliferative effect of AZD4547/BI2536 (Fig 4C) in both H358 and A549 cells. Moreover, siRNA-mediated E2F1 knockdown phenocopied the effects of JNK/p38 dual inhibition, leading to decreased PARP cleavage, dampened proliferation suppression, and compromised apoptosis and G2/M cell-cycle arrest in H358 and A549 cells treated with FGFR1/PLK1 inhibitors (Fig 4D–F; Appendix Fig S4O). Supporting a role for JNK/p38/E2F1 to act downstream of ROS, pharmacological or genetic inhibition of JNK, p38, and E2F1 did not apparently affect the ROS level induced by AZD4547/BI2536 combination in H358 cells (Appendix Fig S4P and Q). Thus, combined PLK1/FGFR1 inhibition abolishes ROS homeostasis, leading to oxidative stress-activated JNK/p38/E2F1 signaling and induction of apoptosis in *KRAS*-mutant lung cancer cells.

## Autophagy protects from FGFR1/PLK1 inhibitor cytotoxicity

Our *in vivo* results showed that, despite potent response to FGFR1/PLK1 inhibitor therapy, *KRAS*-mutant PDX tumors still grew amid the course of treatment (Fig 2F). We also noticed that combined treatment with FGFR/PLK1 inhibitors not only activates p38 (p-p38) and JNK (p-JNK) but also concurrently decreased p-AKT and p-mTOR despite the marginal effect on the ERK1/2 in H358 and A549 cells (Appendix Fig S1F). mTOR negatively regulates autophagy that plays a key role in cancer progression and evolution in response to stressful stimuli (Kim et al, 2011; Yang et al, 2011).

---

**Figure 4. ROS-activated JNK/p38/E2F1 axis contributes to the synergy of FGFR and PLK1 inhibitors.**

A    Immunoblots of H358 and A549 cells treated with vehicle (DMSO), AZD 4547 (5 μM), and BI2536 (5 nM), alone or in combination for 24 h.

B, C    Immunoblots (B) and viability assay (C) of H358 and A549 cells preincubated overnight with SP600125 (2 μM) and SB203580 (1 μM) and subsequently treated with AZD 4547 (5 μM) and BI2536 (5 nM) for 24 h. Data were shown as mean ± SD (n = 3); ***P < 0.001, ****P < 0.0001, and P > 0.05 (ns) by two-way ANOVA with Tukey's multiple comparison test.

D, E    Immunoblots (D) and viability assay (E) of H358 and A549 cells transfected with *E2F1*-specific or control siRNAs followed by treatment (48 h post-transfection) with vehicle (DMSO) or combined AZD 4547 (5 μM)/BI2536(5 nM) for additional 24 h. Data were shown as mean ± SD (n = 3). **P < 0.01, ****P < 0.0001, and P > 0.05 (ns) by two-way ANOVA with Tukey's multiple comparison test.

F    H358 cells transfected with *E2F1*-specific or control siRNAs for 24 h were subsequently treated with vehicle (DMSO), AZD4547 (5 μM), and BI2536 (5 nM), alone or in combination for 48h before apoptotic assay. Data were shown as mean ± SD (n = 3). *P < 0.05, **P < 0.01, and P > 0.05 (ns) by two-way ANOVA with Tukey's multiple comparison test.

Source data are available online for this figure.

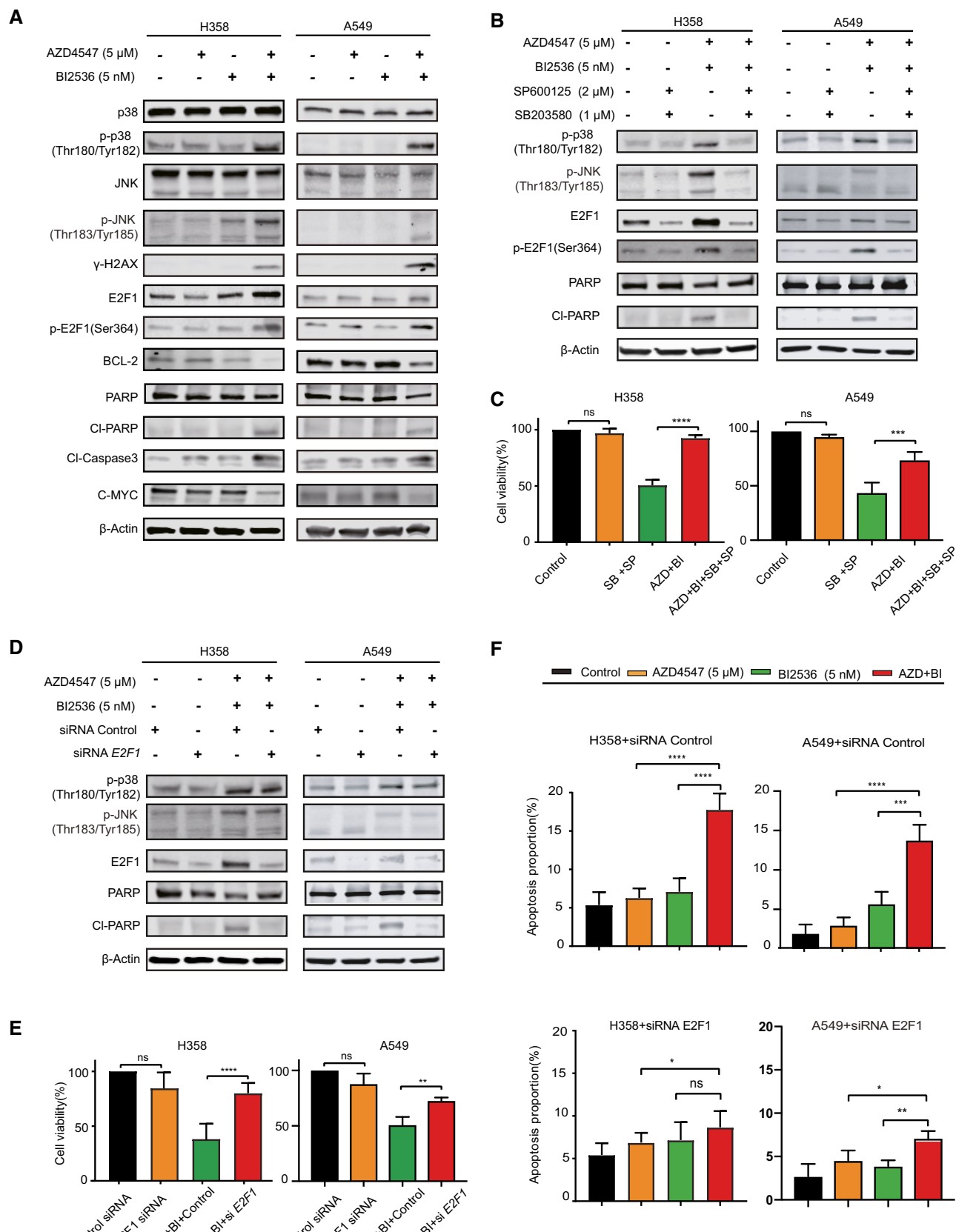

Figure 4.

In light of these observations, we postulated that autophagy might be adaptively activated upon concomitant FGFR/PLK1 inhibition. Supporting this notion, pharmacological FGFR1/PLK1 inhibition led to increased conversion of LC3-I to LC3-II, enhanced degradation of p62, and an elevated ratio of the mCherry:GFP fluorescence, all of which are signs of increased autophagic influx (Parejo *et al*, 2019), in a panel of *KRAS*-mutant cells (Fig 5A–C; Appendix Fig S5A). To further explore the role of autophagy in response to FGFR1/PLK1 inhibitors, H358 and A549 cells were treated with AZD4547/BI2536 in the presence of hydroxychloroquine (HCQ), a clinically approved autophagy inhibitor (Mauthe *et al*, 2018). HCQ at the tested concentration (10 μM) barely impaired the survival of A549 and H358 cells, but strikingly enhanced AZD4547/BI2536 cytotoxicity, leading to more pronounced apoptosis (Fig 5D) and further increased ROS levels (Fig 5E) compared to AZD4547/BI2536 drug combination. Corroborating a role for autophagy in protecting from AZD4547/BI2536-evoked cytotoxicity, genetic depletion of ATG5, a key regulator of autophagy, which, as expected, effectively dampened autophagic influx (decreased LC3-II/I ratio and partial restoration of p62), further enhanced the activity of p38/JNK/E2F1 pathway (increased p-p38, p-JNK, E2F1, and p-E2F1) and promoted apoptosis (CI-PARP) in AZD4547/BI2536-treated H358 cells (Fig 5F). Together, these results indicate that FGFR1/PLK1 inhibitor therapy evokes protective autophagy and that HCQ further increases the synergistic activity of AZD4547/BI2536, which is ascribed, at least in part, to autophagy inhibition.

## A triple combination therapy potently suppresses *KRAS*-mutant lung tumors *in vivo*

Finally, we validated the *in vitro* results in *KRAS*-mutant patient-derived xenograft (PDX) and a genetically engineered mouse model (GEMM) of Kras$^{G12D}$-induced lung adenocarcinoma that precisely recapitulates the human disease (Jackson *et al*, 2001). The triple combination therapy [AZD4547, BI6727 (volasertib) plus HCQ] remarkably attenuated the tumor growth of a PDX (BE564T) and showed durable anti-tumor efficacy (Fig 6A–C), while AZD4547/BI6727-treated tumors still grew despite significantly slower than AZD4547-, BI6727-, HCQ-, or vehicle-treated ones (Fig 6A–C). Accompanied by the greater efficacy, the triple drug regimen was well tolerated in mice (judged by body weights) (Appendix Fig S5B)

and associated with significantly suppressed proliferation (Ki-67) but increased apoptosis (caspase-3) in residual tumors (Fig 6D and E). Investigations with two additional PDXs derived from primary *KRAS*-mutant lung cancer cells (PF563, PF139) had similar results (Fig 6F–K).

Treatment of a GEMM of Kras$^{G12D}$-induced lung adenocarcinoma showed similar results, with only the triple combination therapy producing marked tumor suppression, as indicated by macroscopic examination of tumor lesions and histological evaluation of hematoxylin and eosin (H&E)-stained tissue sections (Fig 7A and B). These observations were subjected to further scrutiny by quantitative measurement, which showed that the lung tumor burden (tumor area/total lung area), tumor size, and tumor number were significantly reduced in mice treated with the triple regimen compared to those with AZD4547/BI6727 (Fig 7C and D). Consistently, the triple drug combination significantly downregulated the tumor grade (Fig 7E) and reduced the proliferative index compared to vehicle and single agents as determined by the percentage of Ki-67-positive cells, although we did not detect a significant difference of Ki-67 positivity by the triple and dual drug regimens (Fig 7F). Notably, the dose of both AZD4547 and volasertib used in our *in vivo* study is far below clinically tolerable doses (Sebastian *et al*, 2010; Zhang *et al*, 2012; Döhner *et al*, 2014) and we did not observe apparent toxicities in the PDX and GEMM tumor models (Appendix Fig S5B and C), consistent with the criteria of a synergistic combination therapy (Lehár *et al*, 2009). Importantly, residual tumors from the triple AZD4547/BI6727/HCQ group showed marked increase in p-JNK compared to those from the AZD4547/BI6727 group (Appendix Fig S5D), and this increase was accompanied by an elevated LC3-II/LC3-I ratio (Appendix Fig S5D), a surrogate marker of decreased autophagosome–lysosome fusion upon HCQ treatment (Mauthe *et al*, 2018). Importantly, AZD4547/BI6727 combination remarkably upregulated ROS in murine KP cells (*KRAS*$^{G12D}$, $p53^{-/-}$) derived from *Kras*-mutant lung adenocarcinoma, and the presence of HCQ further increased AZD4547/BI6727-induced ROS levels in KP cells (Appendix Fig S5E), reinforcing our *in vitro* results.

In summary, our *in vitro* and *in vivo* data demonstrate that FGFR1 and PLK1 inhibitors constitute a synergistic drug combination and that the FDA-approved HCQ further enhances cytotoxicity of the treatment in *KRAS*-mutant lung cancer (Fig 7G).

---

**Figure 5. Autophagy protects from the cytotoxicity of FGFR1/PLK1 inhibitors.**

A  Immunoblots of H358 and A549 cells treated for 24 h with vehicle (DMSO), AZD 4547 (5 μM), and BI2536 (5 nM), alone or in combination.

B  A549 cells stably expressing mCHerry-GFP-LC3B were exposed to vehicle (DMSO), AZD4547 (5 μM), and BI2536 (5 nM), alone or in combination for 24 h. Cell were then fixed and processed for immunofluorescence; scale bar = 10 μm.

C  A549 and H358 cells expressing mCHerry-GFP-LC3B were treated with vehicle (DMSO), AZD 4547 (5 μM), and BI2536 (5 nM), alone or in combination for 24 h before flow cytometry-based analysis for mCherry and GFP (upper). Shown underneath is quantification of the data. **$P < 0.01$ and ***$P < 0.001$ by two-way ANOVA with Tukey's multiple comparison test. Data are shown as mean ± SD (error bar) of three independent experiments ($n = 3$).

D  Apoptotic assay of A549 and H358 cells treated with vehicle (DMSO), AZD 4547 (5 μM), and BI2536 (5 nM) for 48h in the presence or absence of HCQ (10 μM). Data are presented as mean ± SD ($n = 3$). *$P < 0.05$ and ns $P > 0.05$ by two-way ANOVA with Tukey's multiple comparison test.

E  Flow cytometry-based ROS measure in H358, A549, and MIA PaCa-2 cells treated with vehicle (DMSO), AZD4547 (5 μM), and BI2536 (5 nM), alone or in combination, in the presence or absence of HCQ (10 μM) for 24 h. Quantification of relative ROS levels was shown to the right. Data are presented as mean ± SD ($n = 3$). *$P < 0.05$, **$P < 0.01$, ****$P < 0.0001$, and $P > 0.05$ (ns) by two-way ANOVA with Tukey's multiple comparison test.

F  Immunoblots of H358 cells transfected with *ATG5*-specific or control siRNAs for 48 h and before treated for 24 h with vehicle (DMSO) or combined AZD 4547 (5 μM)/ BI2536 (5 nM).

Source data are available online for this figure.

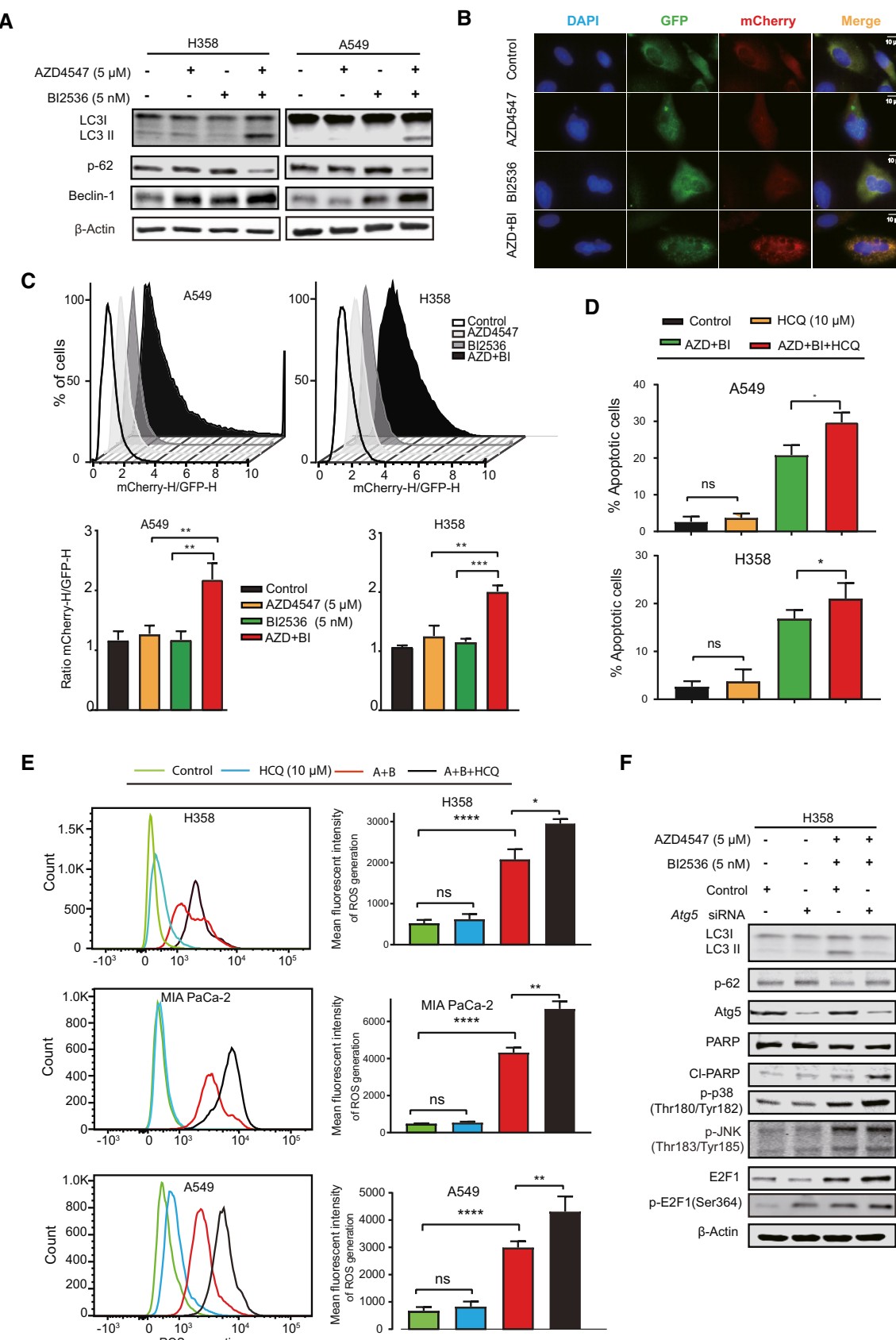

**Figure 5.**

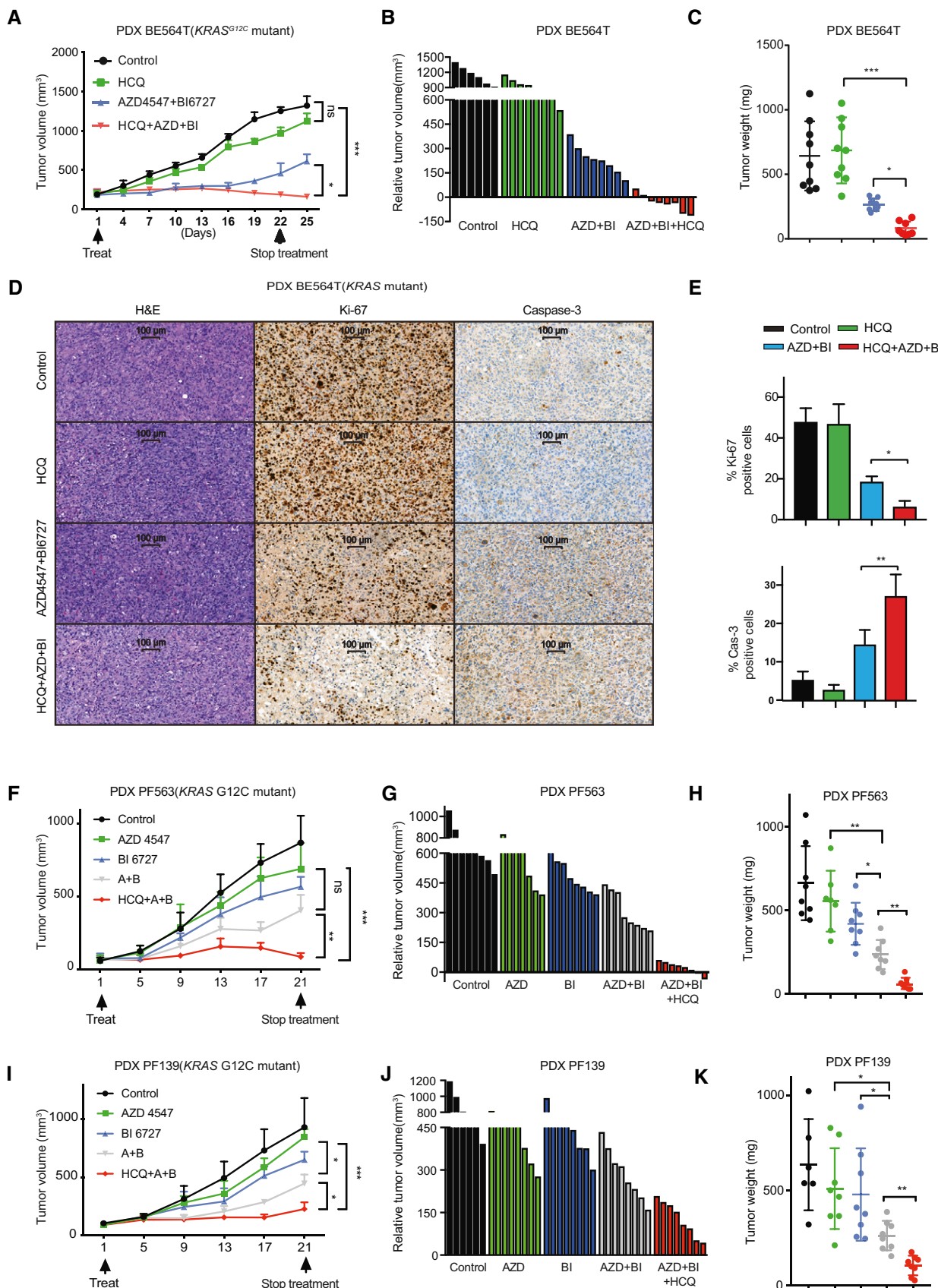

Figure 6.

**Figure 6. A triple combination therapy for *KRAS*-mutant lung cancer.**

A   Growth curve of a *KRAS*-mutant lung cancer PDX (BE564T) treated with vehicle, HCQ (30 mg/kg/day), AZD4547 (10 mg/kg/day), and BI6727 (5 mg/kg/day), alone or in combination. *P < 0.05, ***P < 0.001, and P > 0.05 (ns) by one-way ANOVA with Tukey's multiple comparison test. Data are the mean of tumor volume of each group (5 mice/group); error bar: SD.

B, C   Relative tumor volume (B) and weights (C) of PDX (BE564T) tumors after the treatment. *P < 0.05 and ***P < 0.001 by unpaired two-tailed *t*-test. (C) Data are the mean of tumor weights of each group (5 mice/group); error bar: SD.

D, E   H&E and IHC analysis (Ki-67 and caspase-3) of PDX (BE 564T) tumors after the treatment (D). Quantification of the IHC data for the positivity of Ki-67 and caspase-3 is shown in (E). *P < 0.05 and **P < 0.01 by two-way ANOVA with Tukey's multiple comparison test. Data are shown as mean ± SD (error bar) of three independent experiments (*n* = 3).

F–K   Growth curves (F, I) of *KRAS*-mutant lung cancer PDXs (PF139, PF563) treated with vehicle, HCQ (30 mg/kg/day), AZD4547 (10 mg/kg/day), and BI6727 (5 mg/kg/day), alone or in combination. *P < 0.05, **P < 0.01, and ***P < 0.001 by one-way ANOVA with Tukey's multiple comparison test. Data are the mean of tumor volume of each group (4 mice/group); error bar: SD. Relative tumor volume (G, J) and weights (H, K) of the PDX tumors after the treatment. *P < 0.05, **P < 0.01, ***P < 0.001, and P > 0.05 (ns) by unpaired two-tailed *t*-test. Data are the mean ± SD (error bar) of tumor weights of each group (4 mice/group).

Source data are available online for this figure.

## Discussion

Despite long and persistent efforts, therapeutic targeting of *KRAS*-mutant cancer has remained a significant challenge in clinical oncology. In the present study, we demonstrate that *KRAS*-mutant lung and pancreatic cancers are highly sensitive to concomitant inhibition of FGFR1 and PLK1. This sensitivity arises due to an underlying metabolic vulnerability such that upon FGFR1/PLK1 inhibition, oxidative stress surges due to cytotoxic ROS accumulation, leading to ROS-mediated hyperactivation of the JNK/p38 signaling axis and E2F1-induced apoptosis. We further show that concomitant targeting of FGFR1/PLK1 adaptively evokes autophagy that limits the cytotoxicity of the combination treatment and that abrogation of protective autophagy fully releases the therapeutic potential of FGFR1/PLK inhibitor therapy, leading to greater efficacy and limited toxicity in preclinical cancer models. These results suggest an unanticipated crosstalk, whereby FGFR1 and PLK1 cooperate in surveillance of metabolic ROS homeostasis, which is essential for *KRAS*-induced tumorigenicity. As FGFR and PLK1 inhibitors are under active clinical development (Dieci *et al*, 2013; Gutteridge *et al*, 2016), our findings provide a new rationale of synergistic combination therapy that is readily translatable for patients with *KRAS*-mutant lung cancer.

Our chemical synthetic lethal screens in isogenic cell lines harboring wild-type or mutant alleles of *KRAS* (Langsch *et al*, 2016) reveal a superior combinatorial index of AZD4547 and BI2536 compared to other drug combinations, including the previously reported ROCK inhibitor (Wang *et al*, 2016). The activity and selectivity of FGFR/PLK1 inhibitors as a synergistic combination treatment (Lehár *et al*, 2009), which is restricted to *KRAS*-mutant lung and pancreatic but not colon cancer cells, were validated in multiple *in vitro* and *in vivo* tumor models (e.g., GEMM and PDXs) and further supported by genetic evidence whereby RNAi-based co-depletion of PLK1/FGFR1 drives massive tumor cell death. Notably, the drug dose of FGFR1 and PLK1 inhibitors used in our *in vivo* experiments is below the clinically achievable concentrations (Sebastian *et al*, 2010; Zhang *et al*, 2012), whereby the synergistic anti-tumor activity is not compromised by increased side effects, warranting further clinical investigations of FGFR1/PLK1 inhibitor therapy for treating *KRAS*-mutant cancers.

Aberrant cell metabolism is a hallmark of cancer (Cantor & Sabatini, 2012). Oncogenic KRAS activation has been shown to rewire the metabolic program of cancer cells to fuel the energetic and biosynthetic demands of deregulated proliferation, leading to elevated production of intracellular ROS (Guo *et al*, 2011; Son *et al*, 2013; Yun *et al*, 2015). This increase in ROS is essential to drive the formation and progression of *KRAS*-mutant cancer by upregulating survival and growth factor signaling (Weinberg *et al*, 2010) and generating mitochondrial DNA mutations (Ishikawa *et al*, 2008). However, metabolic ROS is potentially detrimental and a critical issue for cancer cells is to keep ROS at levels beneficial for tumor development but insufficient to induce cell death (Nagel *et al*, 2016; Storz, 2017), which is counterbalanced by anti-oxidant processes such as the transcription factor NRF2 (also known as NFE2L2)-dependent oxidative stress-responsive programs (DeNicola *et al*, 2011). We demonstrate here that FGFR1 cooperates with PLK1 to maintain ROS homeostasis in *KRAS*-mutant lung and pancreatic cells, such that oxidative stress levels overwhelm the anti-oxidant capacity upon FGFR1 and PLK1 inhibition, resulting in apoptosis. Our data suggest an unexpected crosstalk of FGFR1/PLK1 with cellular responses to oxidative stress, and it will be interesting to explore whether and how NRF2 contributes to FGFR1- and PLK1-dependent metabolic surveillance.

JNK and p38 are evolutionarily conserved stress-activated protein kinases that function as critical determinants for cell fate in response to various environmentally stressful signals (Arimoto *et al*, 2008). We identify JNK/p38 as a target of FGFR/PLK1 inhibitor-driven drug synergy, consistent with previous reports showing that stress-activated p38 and JNK induce cell-cycle arrest and apoptosis (Duch *et al*, 2012; Darling & Cook, 2014; Wu *et al*, 2020). Moreover, we demonstrate that elevated ROS/JNK/p38 signaling phosphorylates and activates E2F1 to provoke E2F1-dependent apoptosis, in line with the role of E2F1-activated transcriptional program in apoptosis induction in response to DNA damage (Stevens *et al*, 2003; Lazzerini Denchi & Helin, 2005). Interestingly, *KRAS*-mutant colon cancer cells appear to show a greater dependency on PLK1 for survival than *KRAS*-mutant lung and pancreatic tumor cells, as single treatment with PLK1 inhibitors (e.g., BI2536) is sufficient to induce high levels of ROS and activation of the p38/JNK/E2F1 axis. These observations may explain why the synergy of FGFR and PLK1 inhibitors observed in *KRAS*-mutant lung and pancreatic cancer cells does not translate to *KRAS*-mutant colon cancer cells.

Autophagy has been considered to be a "double-edged sword" in tumorigenesis, which inhibits tumor initiation at an early stage but gets adopted by tumor cells as a survival mechanism at an advanced stage (White, 2012). We identify autophagy as a defense mechanism

in response to FGFR/PLK1 inhibitor-induced surge of metabolic stress, which is consistent with recent studies showing that *KRAS*-driven tumor cells depend on autophagy to help reduce ROS and to provide substrates to fuel cell metabolism (Guo *et al*, 2011; Yang

*et al*, 2011, 2020; Lee *et al*, 2019). Our results are also in line with the finding that autophagy is a mechanism of resistance to MAPK inhibition in *RAS*-driven pancreatic cancers (Kinsey *et al*, 2019; Bryant *et al*, 2020). Further corroborating the antagonistic role of autophagy,

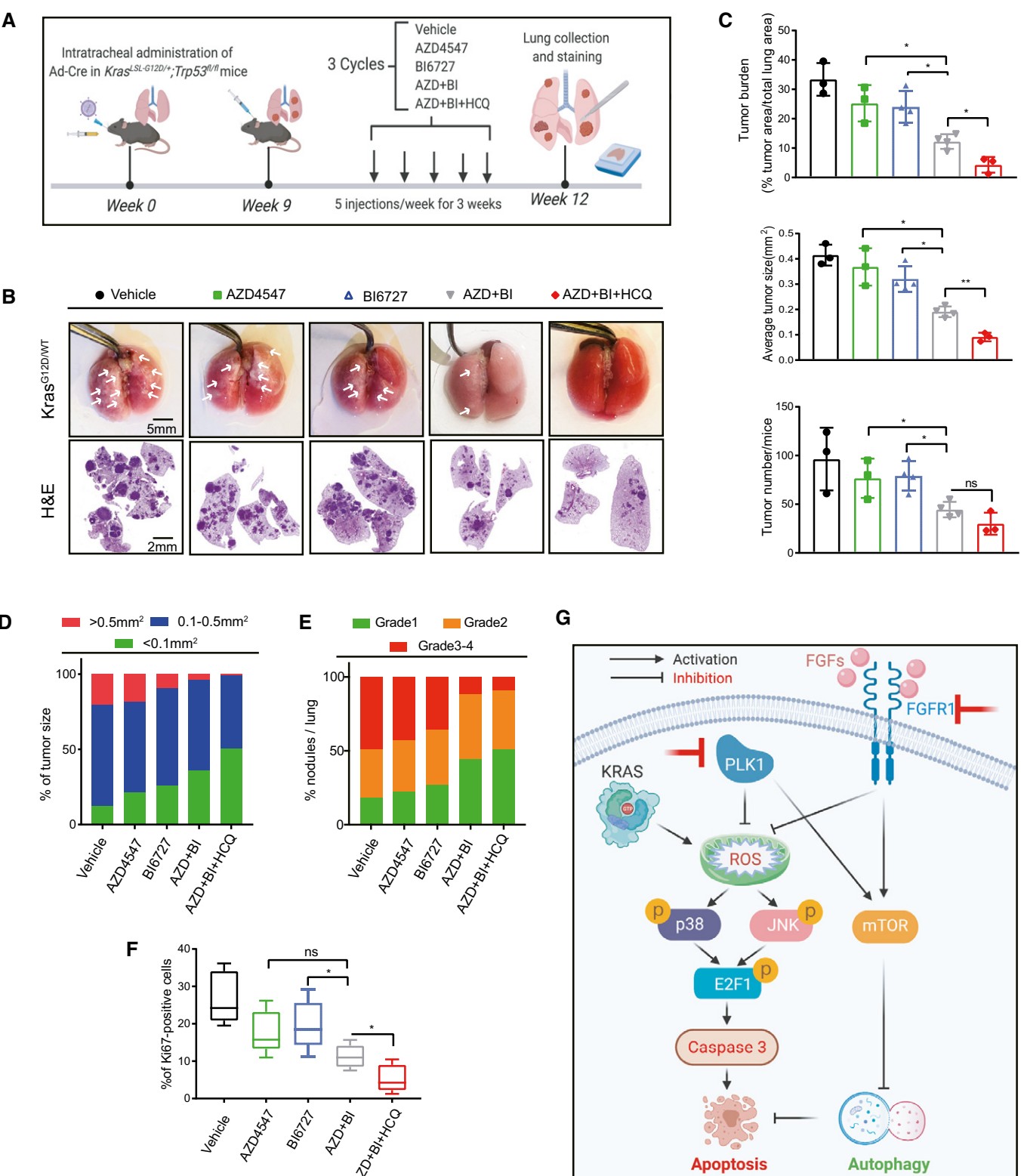

**Figure 7.**

**Figure 7.   *In vivo* efficacy of the combination therapy in *Kras*-induced lung adenocarcinoma.**

A   Schematic of GEMM of *Kras*-induced adenocarcinoma. Established lung tumors in *LSL-Kras*^G12D mice were treated with vehicle, AZD4547 (10 mg/kg/day), BI6727 (5 mg/kg/day), and HCQ (30 mg/kg/day) as indicated.
B   Lung morphology and H&E staining of lung tissue sections after 3-week treatment.
C   Tumor burden, average tumor size, and tumor number (determined by QuPath software) after the treatment. $*P < 0.05$, $**P < 0.01$, and $P > 0.05$ (ns) by two-way ANOVA with Tukey's multiple comparison test. Data are presented as mean $\pm$ SD (error bar) of three or four replicates ($n = 3$ or $4$).
D   The tumor size in different treatment groups, calculated by number of tumors at the indicated size range divided by total tumor number of the treatment group. Data are the mean of four replicates ($n = 4$).
E   Tumor grade analysis (% of nodules/lung) after the treatment. The analysis was based on DuPage *et al* (2009). Data are the mean of three replicates ($n = 3$).
F   Boxplots showing the percentage of Ki-67-positive cells in different treatment groups. Data are based on IHC staining of lung tissue section of *LSL-Kras*^G12D mice. $*P < 0.05$ and $P > 0.05$ (ns) by two-way ANOVA with Tukey's multiple comparison test. The boxplots span from the first to third quartile, and the whiskers extend to 1.5× the interquartile range (IQR). The horizontal line within the boxes represents the median of each group of 10 replicates ($n = 10$).
G   Working model depicting synthetic lethal interaction of FGFR1/PLK1 inhibitors and autophagy as a protective mechanism in *KRAS*-mutant cancer cells.

Source data are available online for this figure.

the clinically approved autophagy inhibitor chloroquine (Mauthe *et al*, 2018) potently enhances the cytotoxicity of FGFR1/PLK1 inhibitors in *KRAS*-mutant lung cancer cells *in vitro* and *in vivo*. Further studies aimed at understanding key regulators of autophagy altered by combined FGFR1/PLK1 inhibition will be necessary.

In summary, we reveal that PLK1 blockage potentiates FGFR-targeted therapy by activating the ROS/JNK/p38/E2F1 signaling axis and inducing apoptosis. Greater efficacy can be further achieved by therapeutic targeting of FGFR/PLK1 plus HCQ therapy. Our data suggest a synergistic combination therapy for *KRAS*-mutant lung and pancreatic cancer, which is also worthwhile to be tested in other *KRAS*-driven malignancies.

# Materials and Methods

## Cell culture and reagents

All cells used in this study are listed in Appendix Table S3. Patient-derived primary *KRAS*-mutant LUAD cells (PF139, PF563) were established as we recently reported (Stockhammer *et al*, 2020). Human bronchial epithelial cells BEAS-2B and the isogenic counterparts BEAS-2B-KRAS (expressing KRAS^G12V) were described in our previous study (Langsch *et al*, 2016). All cell lines have been authenticated by DNA fingerprinting using highly polymorphic short tandem repeat (STR) analysis and free from mycoplasma contamination (Microsynth, Bern, Switzerland). Cells were cultured in RPMI-1640 supplemented with 10% fetal bovine serum/FBS (Cat. #10270-106; Life Technologies, Grand Island, NY, USA) and 1% penicillin/ streptomycin solution (Cat. #P0781, Sigma-Aldrich) at 37°C with 95% air/5% $CO_2$. All inhibitors were purchased from Selleckchem (Houston, TX, USA) (Appendix Table S4).

## Cell viability and clonogenic survival assay

Tumor cells were seeded in 96-well plates (2,500 cells/well) and were treated next day with various drugs for 72 h unless otherwise indicated. Cell viability was determined by PrestoBlue Cell Viability Reagent (Thermo Fisher Scientific), following the manufacturer's instructions. The PB reagent was added into media directly (1:10 dilution) and incubated for 20 min-2 h, and the fluorescence was read (excitation 570 nm; emission 600 nm) at recommended time of incubation. The efficacy of drugs on cell growth was normalized to

untreated control. Each data point was generated in triplicate, and each experiment was done three times ($n = 3$). Best-fit curve was generated in GraphPad Prism [(log (inhibitor) versus response (-variable slope four parameters)]. Error bars are mean $\pm$ SD. The combination index (CI) was calculated by ComboSyn software (ComboSyn Inc., http://www.combosyn.com/).

Clonogenic assay was done as described (Liang *et al*, 2019; Yang *et al*, 2019b). In brief, tumor cells seeded in 6-well plates (1,000–5,000 cells/well) were treated with drugs for 1 day and cultured in the absence of drugs for 2–3 weeks, and the resulting colonies were stained with crystal violet (0.5% dissolved in 25% methanol).

## Small interfering RNA (siRNA) transfection

Cells were seeded into 6-well plate at a density of 0.3 $\times 10^6$ cells per well and allowed to reach approximately 70% confluence on the day of transfection. Cells were transfected with 20 nM siRNA using SiTran1.0 (TT300001; OriGene Technologies) according to the manufacturer's protocol. *FGFR1*, *FGFR2*, *FGFR3*, *PLK1*, *ATG5*, and *E2F1* knocked down by specific pooled siRNA duplexes (SR320159, SR301586, SR301584, SR321347, SR322789, and SR305006, OriGene Technologies, Rockville, MD, USA). Negative Control siRNA Duplex (Cat. # SR30004, OriGene Technologies) was used as control.

## Immunoblotting, immunohistochemistry, and immunofluorescence

Cell lysates were prepared and Western blot analysis was performed as described (Liang *et al*, 2019; Yang *et al*, 2019b, 2021). In brief, equal amounts of protein lysates were resolved by SDS–PAGE (Cat. #4561033; Bio-Rad Laboratories, Hercules, CA, USA) and transferred onto nitrocellulose membranes (Cat. #170-4158; Bio-Rad). Membranes were then blocked with blocking buffer (Cat. #927-4000; Li-COR Biosciences, Bad Homburg, Germany) for 1 h at room temperature (RT) and incubated with appropriate primary antibodies overnight at 4°C (Appendix Table S5). IRDye 680LT-conjugated goat anti-mouse IgG (Cat. #926-68020) and IRDye 800CW-conjugated goat anti-rabbit IgG (Cat. #926-32211) from Li-COR Biosciences were used at 1:5,000 dilutions. Finally, signals of membrane-bound secondary antibodies were imaged using the Odyssey Infrared Imaging System (Li-COR Biosciences).

For immunofluorescence, tumor cells grown on poly-lysine-treated coverslides were fixed with 4% paraformaldehyde for

15 min at RT and permeabilized with cold methanol ($-20°C$) for 5 min or with 0.1% Triton X-100/PBS at RT for 15 min before incubated overnight at 4°C with primary antibodies (Appendix Table S5). The cells were incubated for 1 h at RT with Alexa Fluor 647 goat anti-mouse IgG (Cat. #A21236) or Alexa Fluor 488 goat anti-Rabbit IgG (Cat. #A11034) from Invitrogen (Eugene, OR, USA). Nuclei were counterstained by 4′,6-diamidino-2-phenylindole. Images were acquired on a ZEISS Axioplan 2 imaging microscope (Carl Zeiss MicroImaging, Göttingen, Germany) and processed using Adobe Photoshop CS6 v.13 (Adobe Systems, San Jose, CA, USA).

For immunohistochemical studies, surgically removed xenograft tumors were formalin-fixed and paraffin-embedded (FFPE) and stained with hematoxylin and eosin (H&E) using standard protocols. FFPE tissue blocks were sectioned at 4 μm, deparaffinized, rehydrated, and subsequently stained with appropriate antibodies (Appendix Table S5) using the automated system BOND RX (Leica Biosystems, Newcastle, UK). Visualization was performed using the Bond Polymer Refine Detection kit (Leica Biosystems) as instructed by the manufacturer. Images were acquired and processed using Adobe Photoshop CS6 v13 (Adobe Systems Incorporated).

### Cell cycle analysis

Lung cancer cells were treated for 24 h with vehicle or the indicated drugs. After treatment, cells were harvested, fixed with 75% ethanol at $-20°C$ overnight, and stained with propidium iodide (PI)/RNase staining buffer for 15 min. Flow cytometry analysis was performed on a BD Biosciences LSRII flow cytometer. Three independent experiments were performed.

### Quantitative real-time PCR (qRT–PCR)

Total RNA was isolated and purified using RNeasy Mini Kit (74106; Qiagen, Hilden, Germ). Complementary DNA (cDNA) was synthesized by the High capacity cDNA reverse transcription kit (4368814; Applied Biosystems, Foster City, CA, USA) according to the manufacturer's instructions. Real-time PCR was performed in triplicate on a 7500 Fast Real-Time PCR System (Applied Biosystems) using TapMan primer/probes (Applied Biosystems): *FGFR1*, Hs00241111_m1; *FGFR2*, Hs01552918_m1; and *FGFR3*, Hs00179829_m1. *ACTB*, Hs01060665_g1 was used as endogenous normalization controls.

### Apoptosis assays

Lung cancer cells were treated for 48 h with vehicle or the indicated drugs. After treatment, cells in the supernatant and adherent to plates were collected, washed with PBS and pooled before suspended in 400 μl binding buffer and stained with the Annexin V Apoptosis Detection Kit -FITC (Cat. #88-8005; Thermo Fisher Scientific, Waltham, MA, USA) according to the manufacturer's instructions. Flow cytometry analysis was performed on a BD Biosciences LSRII flow cytometer. Three independent experiments were performed.

### Measurement of ROS generation

Intracellular ROS production was detected using Reactive Oxygen Species (ROS) Detection Assay Kit (Cat. # K936-100; BioVision,

Milpitas, CA, USA). Cells were plated in six-well plates at a density of $0.3 \times 10^6$ cells per well for overnight. Cells were collected and washed with ROS generation buffer and then incubated with H2DCFDA at 37°C in the dark for 60 min. Then, cells were treated with indicated reagents and the ROS level was determined by flow cytometry (BD Biosciences, San Jose, CA, USA).

### Autophagic flux assay using mCherry-eGFP-LC3B

Lentiviral vectors (pMD2GVSGV, pMDlg/pRRE, and pRSV-rev) are used to pack and deliver mCherry-eGFP-LC3B and a puromycin resistance gene to H358 and A549 as described (Parejo et al, 2019). Selection in puromycin is conducted for 3–5 days starting 2 days post-infection. Images were acquired on a ZEISS Axioplan 2 imaging microscope (Carl Zeiss MicroImaging, Göttingen, Germany) and processed using Adobe Photoshop CS6 v.13 (Adobe Systems, San Jose, CA, USA). Rationmetrics of mCherry/eGFP was performed on a BD Biosciences LSRII flow cytometer.

### Patient samples

Surgically resected tumor specimens from a lung cancer patient (BE564T, 67-year-old male, $KRAS^{G12C}$-mutant lung adenocarcinoma) were obtained from Lung Cancer Center (LCC), Bern University Hospital. *KRAS* mutations in NSCLC tumors were analyzed by pyrosequencing or Sanger sequencing as we previously described (Langsch et al, 2016). PF563 ($KRAS^{G12C}$ lung adenocarcinoma) and PF139 ($KRAS^{G12C}$ lung adenocarcinoma) were derived from a 67-year-old female patient and a 75-year-old male patient, respectively. The establishment of PF139 and PF563 cells was approved by the Ethics Committee at the University Hospital Essen (#18-8208-BO), and written consents were obtained from the patients. Authentication was performed by SNP-based cell identification (Multiplexion, Heidelberg, Germany). All human studies were performed under the auspices of protocols approved by the institutional review board (KEK number: 042/15 and 200/2014). Informed consent was obtained from all human subjects and that the experiments conformed to the principles set out in the WMA Declaration of Helsinki and the Department of Health and Human Services Belmont Report.

### *In vivo* mouse study

Mouse studies were conducted in accordance with Institutional Animal Care and Ethical Committee-approved animal guidelines and protocols. Age- and gender-matched NSG (NOD-*scid IL2Rγ^{null}*) mice were used for human cancer cell line induced xenografts and patient-derived xenografts (PDXs). When tumors were palpable, mice were randomly assigned to treatment groups: (i) control; (ii) BI6727 (5 mg/kg, i.p., once daily); (iii) AZD4547 (10 mg/kg, p.o., once daily); (iv) drug combination of AZD4547 and BI6727; and (v) triple combination [AZD4547, BI6727 plus HCQ (30 mg/kg/per day)]. Tumor size was measured by caliper every 3 days. Tumor volume was calculated as follows: $(length \times width^2)/2$.

For GEMM, $Kras^{LSL-G12D/+}$ and $Trp53^{fl/fl}$ mice (8–12 weeks old) were anaesthetized by intraperitoneal injection of ketamine (80 mg/kg body weight) and xylazine (10 mg/kg body weight) and infected intratracheally with a dose of $2.5 \times 10^7$ PFU in a total volume of

**The paper explained**

**Problem**

Despite the overwhelming prevalence of oncogenic *KRAS* mutations in human cancers and the continuity of tremendous efforts, effective treatment of *KRAS*-mutant cancers has remained an overarching challenge in clinical oncology.

**Results**

Our synthetic lethal chemical screens identified fibroblast growth factor receptor 1 (FGFR1) as a target to promote the anti-cancer efficacy of polo-like kinase 1 (PLK1) inhibitor therapy in *KRAS*-mutant lung and pancreatic cancer. The synergistic cytotoxicity of PLK1 and FGFR1 inhibitors converges at a metabolic liability that offsets oxidative stress associated with oncogenic *KRAS* mutations and leads to activation of JNK/p38 signaling and E2F1-induced apoptosis. We also uncovered a compensatory mechanism by which autophagy protects against the combination of PLK1 and FGFR inhibitors and validated a triple modality that potently suppresses the growth of *KRAS*-mutant lung tumors in multiple mouse models, including patient-derived xenografts and GEM model of *Kras*-induced lung adenocarcinoma.

**Impact**

KRAS epitomizes an oncoprotein that is notoriously recalcitrant to targeted therapies. The identification of a synergistic drug combination and a therapeutically exploitable protective mechanism to enhance efficacy suggests a new rationale for the treatment of *KRAS*-mutant cancers.

75 μl of adeno-Cre per mouse as we previously described (Saliakoura *et al*, 2020). Eight weeks after virus inhalation, animal were randomized into five groups: (i) control; (ii) BI6727 (5 mg/kg, i.p., once daily); (iii) AZD4547 (10 mg/kg, p.o., once daily); (iv) combination of AZD4547 and BI6727; and (v) triple combination [AZD4547, BI6727 plus HCQ (30 mg/kg/day)]. The treatment lasted for 3 weeks. Tumor burden, average tumor size, and tumor number were calculated by using QuPath software. Tumor grade was assessed according to the previous study (DuPage *et al*, 2009).

**Statistical analysis**

Statistical analyses were performed using GraphPad Prism 7.01 (GraphPad Software Inc.) unless otherwise indicated. All samples that met proper experimental conditions were included in the analysis, and sample size was not pre-determined by statistical methods but rather based on preliminary experiments. Group allocation was performed randomly. In all studies, data represent biological replicates (n) and are depicted as mean values ± SD or mean values ± SEM as indicated in the figure legends. Comparison of mean values was conducted with unpaired, two-tailed Student's *t*-test, one-way ANOVA, or two-way ANOVA with Tukey's multiple comparison test as indicated in the figure legends. In all analyses, $P < 0.05$ were considered statistically significant.

# Data availability

This study includes no data deposited in external repositories.

**Expanded View** for this article is available online.

## Acknowledgements

We acknowledge Matteo Rossi Sebastiano (Institute of Pharmacology, University of Bern) for animal studies and *in vitro* experiments. The PF139 and PF563 cell lines were established in collaboration with the West-German Biobank Essen (WBE). This work was supported by grants from Swiss National Science Foundation (SNSF; #310030_192648; to R-W. Peng), Swiss Cancer League/ Swiss Cancer Research Foundation (#KFS-4851-08-2019; to R-W. Peng), Cancer League of the Canton of Bern (to R-W. Peng; G. J. Kocher.), and PhD fellowships from China Scholarship Council (to Z.Y., H.Y., Y. G., H.D.).

## Author contributions

ZY and S-QL designed and performed the experiments and analyzed data. MS, HY, EV, and GK performed experiments and analyzed data. MT and BH provided key reagents and technical support. LZ, YG, DX, HD, and TMM helped with data collection and analysis. GJK, WW, and RAS provided clinical samples. GJK and RAS provided financial support. R-WP conceived the project, supervised the study, and wrote the paper with inputs from all co-authors.

## Conflict of interest

The authors declare that they have no conflict of interest.

## For more information

i.   TCGA: https://portal.gdc.cancer.gov/
ii.  GSEA: http://software.broadinstitute.org/gsea/index.jsp
iii. GSE73024: https://www.ncbi.nlm.nih.gov/geo/query/acc.cgi?acc= GSE73024
iv.  cBioportal: https://www.cbioportal.org/

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
