## [Review Process File · EMBO Molecular Medicine]

Synergistic effects of FGFR1 and PLK1 inhibitors target a metabolic liability in *KRAS*-mutant cancer

Zhang Yang, Shun-Qing Liang, Maria Saliakoura, Haitang Yang, Eric Vassella, Georgia Konstantinidou, Mario Tschan, Balazs Hegedüs, Liang Zhao, Yanyun Gao, Duo Xu, Haibin Deng, Thomas M. Marti, Gregor J. Kocher, Wenxiang Wang, Ralph A. Schmid, Ren-Wang Peng
DOI: 10.15252/emmm.202013193

Corresponding authors: Ren-Wang Peng (renwang.peng@insel.ch) , Ralph A. Schmid (Ralph.Schmid@insel.ch)

Review Timeline:

Submission Date:	2nd Aug 20
Editorial Decision:	26th Aug 20
Revision Received:	26th May 21
Editorial Decision:	17th Jun 21
Revision Received:	6th Jul 21
Accepted:	9th Jul 21

Editor: Zeljko Durdevic

Transaction Report:

Dear Dr. Peng,

Thank you for the submission of your manuscript to EMBO Molecular Medicine. We have now received feedback from the three reviewers who agreed to evaluate your manuscript. As you will see from the reports below, the referees acknowledge the interest of the study but also raise some concerns that should be addressed in a major revision.

Addressing the reviewers' concerns in full will be necessary for further considering the manuscript in our journal, and acceptance of the manuscript will entail a second round of review. EMBO Molecular Medicine encourages a single round of revision only and therefore, acceptance or rejection of the manuscript will depend on the completeness of your responses included in the next, final version of the manuscript. For this reason, and to save you from any frustrations in the end, I would strongly advise against returning an incomplete revision.

We realize that the current situation is exceptional on the account of the COVID-19/SARS-CoV-2 pandemic. Therefore, please let us know if you need more than three months to revise the manuscript.

I look forward to receiving your revised manuscript.

***** Reviewer's comments *****

Referee #1 (Remarks for Author):

Ref: EMM-2020-13193

The manuscript entitled "Synergistic effects of FGFR1 and PLK1 inhibitors target a metabolic liability in KRAS-mutant cancer" by Zhang Yang, Shun-Qing Liang et al describes combined FGFR1 and polo-like kinase 1 (PLK1) treatment as an effective therapeutic strategy against KRAS-driven tumors both in vitro and in vivo. Interestingly, anti-proliferative effects are exclusively observed in lung and pancreatic cancer cells but not in colon nor KRAS-wild-type cancer cells.

From a mechanistic point of view, the therapeutic effect is exerted through ROS-dependent activation of JNK/p38 pathway and E2F1-induced apoptosis. Finally, the authors unraveled the role of autophagy in protecting from PLK1/FGFR1 inhibitor cytotoxicity and provided in vitro and in vivo data on triple chloroquine/FGFR1/PLK1 treatment resulting in potent and durable responses in KRAS-driven LUAD.

The study addresses an important topic, as new therapeutic targets are clearly needed in KRAS-

mutant tumors (particularly for KRAS-driven LUAD). Although some in vitro experimental aspects are not fully dissected, in vivo data are convincing and well presented.

Overall, this study describing a new strategy to treat KRAS-driven LUAD is novel and of potential relevance to EMBO Molecular Medicine. Nevertheless, significant revisions are required to address the points detailed below:

- 1) The initial drug screening in BEAS-2B cells is supposed to be done in what are described as "isogenic" cells. However, according to the referenced study (Langsch et al, 2016), KRAS G12V has been introduced by retroviral transduction, using non-endogenous promoter and with unknown number of integrated copies. Actually, total KRAS protein levels seem to be higher in BEAS-2B-KRAS compared to control (Fig. S1). I wonder whether the same screening with the proper control-BEAS-2B cells transduced with wild-type KRAS in the same retroviral vector- would return different (or more) hits.
- 2) All the in vitro validation is performed using very high drug concentrations and only using KRAS-mutant sensitive cells: proper negative controls using for example EGFR-mutant cell lines and/or KRAS-mutant colon cancer cell lines are missing. These controls would help understanding at the molecular level why FGFR1/PLK1 inhibition is selectively effective in KRAS-mutant LUAD and PDAC but not EGFR- or BRAF-mutant LUAD or KRAS-mutant CRC.
- 3) Table S1 would be more informative with another column reporting drugs IC50 in BEAS-2B cells as well.
- 4) Basal levels of FGFR1 detected by western blot are very faint (Fig. S3A) and make the interpretation of siRNA knock-down difficult. mRNA levels should be presented in parallel.
- 5) In colon cancer cell lines, PLK1 knock-down alone already exerted a very powerful therapeutic effect. It would be useful to have a table with IC50 (single agent) for all the cell lines tested in this study.
- 6) PDX number should be increased up to n=3 for sake of statistical significance (especially in light of the extreme heterogeneity of KRAS-mutant LUAD).
- 7) "We noticed that genetic depletion of FGFR1 increased the expression of PLK1 and vice versa (Figure S3A)". Based on the panels presented here, this seems to be an overstatement. Proper western blot quantification should be presented in parallel.
- 8) TCGA analysis on a cohort of KRAS-mutant LUAD (n=141) revealed a negative correlation between PLK1 mRNA and FGFs mRNA. Does this stand true also in PDAC? What about EGFR or BRAF-mutant LUAD? Or KRAS-mutant CRC?
- 9) mTORC suppression upon treatment with CH5183284 should be confirmed by siRNA against FGFR1 (Fig. S3).
- 10) Western blot analysis of PDX samples (Fig. S4D) is poor and quality should be improved. Also, full proteins panel (as in Fig. 4A) should be presented.
- 11) Are ROS levels affected by treatment with SP600125 or SB203580? And by E2F1 knock-down?
- 12) In vivo data presented in Fig. 6 are impressive. It would be very informative to have residual tumors analyzed for autophagy markers, ROS levels, p38 pathway, E2F1 levels.
- 13) Panel S3E should be cited in the text before S4A.

Referee #2 (Remarks for Author):

In the current manuscript, the authors identify a novel synergistic combination of FGFR1 and PLK1 inhibitors which was effective KRAS mutant NSCLC and PDAC but not CRC. They demonstrated that the combination was effective in vitro and in vivo. The combination lead to increases apoptosis and ROS production as well activation/induction of JNK/p38 and E2F1. Furthermore, they

demonstrated that autophagy plays a protective role in the presence of FGFR1 and PLK1 inhibition and the combination of FGFR1/PLK1 with autophagy inhibition was the most efficacious in vitro and in vivo. In summary, this is a well-written, logical and well-performed study that has identified a novel synergistic combination that is effective against KRAS mutant NSCLC and PDAC. This manuscript could be improved if the mostly minor concerns below are addressed.

1. The authors demonstrate that multiple processes that may be necessary for the combination to inhibit growth including ROS, apoptosis and G2/M. It would be helpful to define the requirement of apoptosis though testing the combination in the presence or absence of a caspase inhibitor. Furthermore, it would be helpful to see induction of ROS, JNK/p38, E2F1, G2/M arrest and apoptosis occur in the KRAS mutant CRC lines for which the combination does not cause growth inhibition.
2. The experiments in Fig 4B-F should be verified in a 2nd KRAS mutant cell line.

Minor

3. The authors state that NAC decreased cl-PARP in H358 and A549 cell lines but the data presented in Fig. 3E is unconvincing for A549.
4. A short discussion in the DISCUSSION section of why KRAS mutant CRC is not responsive to the combination would be helpful.
5. The authors should describe the histology and cancer subtype of the PDX used in this study both in the results section and methods.

Referee #3 (Remarks for Author):

In this study, the authors investigated drug combinations that could lead to more efficacious treatment for KRAS mutant lung cancer cells. Building on previous synthetic lethal studies identifying KRAS mutant cells are more sensitive to PLK1 inhibitors, the authors carried out a small drug combination screen using 21 compounds and identified the FGFR inhibitor AZD4547 to exhibit combination synergy with the PLK1 inhibitor BI2536 in KRAS mutant lung cancer cell lines in vitro and in KRAS mutant lung tumor models in vivo. Mechanistically, combined FGFR and PLK1 inhibition leads to elevated ROS production and autophagy, and inhibition of autophagy using HCQ could further enhance cell killing in a triple-drug combination. Single agent PLK1 inhibitors have met with limited success in clinical trials thus far, and this study presented pre-clinical evidence for a new combination therapy with translational potential.

Major points

1. For Figure 2, the combination synergy experiment in the cell line panel was carried out using a single concentration of AZD4547 (5uM) and BI2536 (5nM). This raises the question of whether at the concentration used these inhibitors are effectively blocking their targets in different cell lines, since presumably the IC50s of these drugs are different amongst different cell lines. It is thus possible that, for KRAS mutant colorectal cell lines and KRAS WT cell lines, the lack of synergy could be due to lack of target inhibition at the chosen drug concentrations. The authors should present evidence that AZD4547 is inhibiting FGFR signaling (for examples, by pFGFR and pFRS2 blots), and BI2536 is inhibiting PLK1 activity (for example by using pPLK1 blot) in representative cell lines from this panel to rule out this possibility.
2. Could NAC rescue the toxicity of the triple combination of AZD, BI and HCQ?

Minor points

1. Does FGFR1/2/3 or FRS2 expression differ between KRAS mutant and KRAS WT cell lines?
2. Figure 2E. A WB confirming FGFR1 and PLK1 siRNA knockdown should be shown.
3. Figure 2F. A WB confirming FGFR1 signaling inhibition in PDX BE564T should be shown.
4. Figure 3A. for the GSEA analysis, Why was a different FGFR inhibitor (CH5183284) used and why

were H1581 and H520 cells used? These two cell lines did not appear in the cell line panel study in Figure 2. Does the GFGRi+PLK1i combination exhibit drug synergy in these two cell lines?

5. Figure 5F. ATG5 siRNA knockdown efficiency appears to be poor.

6. Figure 6E. Do the double and triple drug combinations result in a difference in tumor grade in the KP model compared to single agents or control?

Dear Reviewer Experts,

Thank you for your review of our manuscript (EMM-2020-13193) and your constructive and insightful comments and suggestions. With the support of the Editor, we were granted considerable additional time that allowed us to fully address the important points you raised and revise the manuscript accordingly. In particular, we have performed new experiments and provided additional evidence to support our results and conclusions.

Below are our point-by-point responses, with our answers marked in blue.

Referee #1 (Remarks for Author):

The manuscript entitled "Synergistic effects of FGFR1 and PLK1 inhibitors target a metabolic liability in KRAS-mutant cancer" by Zhang Yang, Shun-Qing Liang et al describes combined FGFR1 and polo-like kinase 1 (PLK1) treatment as an effective therapeutic strategy against KRAS-driven tumors both in vitro and in vivo. Interestingly, anti-proliferative effects are exclusively observed in lung and pancreatic cancer cells but not in colon nor KRAS-wild-type cancer cells.

From a mechanistic point of view, the therapeutic effect is exerted through ROS-dependent activation of JNK/p38 pathway and E2F1-induced apoptosis. Finally, the authors unraveled the role of autophagy in protecting from PLK1/FGFR1 inhibitor cytotoxicity and provided in vitro and in vivo data on triple chloroquine/FGFR1/PLK1 treatment resulting in potent and durable responses in KRAS-driven LUAD.

The study addresses an important topic, as new therapeutic targets are clearly needed in KRAS-mutant tumors (particularly for KRAS-driven LUAD). Although some in vitro experimental aspects are not fully dissected, in vivo data are convincing and well presented.

Overall, this study describing a new strategy to treat KRAS-driven LUAD is novel and of potential relevance to EMBO Molecular Medicine.

We thank the reviewer for the coherent summary and positive evaluation of our study.

Nevertheless, significant revisions are required to address the points detailed below:

- 1) The initial drug screening in BEAS-2B cells is supposed to be done in what are described as "isogenic" cells. However, according to the referenced study (Langsch et al, 2016), KRAS G12V has been introduced by retroviral transduction, using non-endogenous promoter and with unknown number of integrated copies. Actually, total KRAS protein levels seem to be higher in BEAS-2B-KRAS compared to control (Fig. S1).

I wonder whether the same screening with the proper control- BEAS-2B cells transduced with wild-type KRAS in the same retroviral vector- would return different (or more) hits.

We thank the reviewer for the insightful comment and fully agree that BEAS-2B cells transduced with the same retroviral vector expressing wild-type *KRAS* might return different (or more) hits.

However, as we reported previously (Langsch et al, 2016), retroviral expression of *KRAS*^{G12V} in BEAS-2B cells results in an oncogenic state of the cells (BEAS-2B-KRAS), e.g., activation of MAPK signaling and increased proliferation compared with BEAS-2B cells, which has allowed us to identify miRNAs specifically upregulated by mutant *KRAS*. In the present study, we performed synthetic lethal chemical screens using the same system, which has enabled the identification of the FGFR1 inhibitor AZD4547 as a novel candidate that synergistically enhances the efficacy of PLK1 inhibitor therapy in *KRAS*-mutant cancer cells. Importantly, our screen also returned ROCK/PLK1 inhibitors as a synergistic drug pair, which had been reported by an independent study (Wang et al, 2016). The recovery of known synergistic drug combination confirmed the credibility of the screening platform and validated the replicability of the screening results.

Notably, it is not uncommon for functional screens to use strategies similar to those we have used. For example, in the study by Wang et al. (2016), synthetic lethal chemical screens were performed in immortalized human ovarian epithelial cells (T29) and their isogenic counterpart generated by retroviral transduction of a *KRAS*^{G12V} expressing construct (Liu et al, 2004).

Langsch S, Baumgartner U, Haemmig S, Schlup C, Schäfer SC, Berezowska S, Rieger G, Dorn P, Tschan MP, Vassella E. miR-29b Mediates NF-κB Signaling in KRAS-Induced Non-Small Cell Lung Cancers. *Cancer Res* 2016; 76: 4160-4169

Liu J, Yang G, Thompson-Lanza JA, Glassman A, Hayes K, Patterson A, Marquez RT, Auersperg N, Yu Y, Hahn WC, Mills GB, Bast RC Jr. A genetically defined model for human ovarian cancer. *Cancer Res.* 2004; 64:1655-63.

Wang J, Hu K, Guo J, Cheng F, Lv J, Jiang W, Lu W, Liu J, Pang X, Liu M. Suppression of KRas-mutant cancer through the combined inhibition of KRAS with PLK1 and ROCK. *Nat Commun* 2016; 7:11363

2) All the *in vitro* validation is performed using very high drug concentrations and only using KRAS-mutant sensitive cells: proper negative controls using for example EGFR-mutant cell lines and/or KRAS-mutant colon cancer cell lines are missing. These controls would help understanding at the molecular level why FGFR1/PLK1 inhibition is selectively effective in KRAS-mutant LUAD and PDAC but not EGFR- or BRAF-mutant LUAD or KRAS-mutant CRC.

We thank the reviewer for the important comments. In the revised manuscript, we have included additional experimental data to address these points.

First, as showed in original submission (**Figure EV1F, EV4B** in the revised manuscript), in KRAS-mutant lung cancer cells (A549, H358), AZD4547 (FGFR inhibitor) in the range of 2.5 μM - 5 μM effectively inhibited p-AKT (S473), a downstream effector of FGFR1 signaling, and 5 nM BI2536 (PLK1 inhibitor) successfully blocked p-PLK1.

Our new Western blot of A549 and H358 cells treated with AZD4546 and BI2536 at different doses (**Figure EV1D, EV1E**) showed that AZD4547 should be used in the range of 1 μM – 10 μM to effectively inhibit p-FRS (substrate of FGFR1) in A549 cells. In H358 cells expressing low levels of FRS2, 1 μM – 10 μM AZD4547 was also required to inhibit p-AKT (S473) (**Figure EV1D**). For BI2536, the lowest dose that efficiently inhibited p-PLK1 in A549 and H358 cells was approximately 5 nM (**Figure EV1E**). These results, which validate the drug doses used our *in vitro* analyses, were described on page 7 of the revised manuscript.

Secondly, we performed *in vitro* validation using additional cancer cell lines, including KRAS-mutant colon (SW620, DLD-1) and KRAS-wild type lung [EBC-1 (EGFR-mutant), H1993 (*c-MET*-amplified)] cancer cells. Results from our new experiments confirmed the synergy between AZD4547 and BI2536 in KRAS-mutant lung (H358, H441, A549) and pancreatic (MIAPaCa-2), but not in SW620, DLD-1, nor in EBC-1 or H1993 cells (**Figure EV2C, D**). These results are consistent with our *in vitro* data shown in **Figure 2A-2E**, and are described on page 7 and 8 of the revised manuscript.

Thirdly, our new experiments with SW620 and EBC-1 cells showed the combination of AZD4547 (5 μ M) and BI2536 (5 nM), despite inhibition of their targets in the cells, did not significantly (or only slightly) induce ROS levels (**Figure EV4F**) and failed to increase the expression of p-JNK, p-p38, p-E2F1, and CI-PARP compared with single agents or vehicle control (**Figure EV4I**), in sharp contrast to the scenario observed in *KRAS*-mutant lung and pancreatic cancer cells (**Figure 3B, 3E, 4A**). These observations provide a plausible explanation why FGFR1/PLK1 inhibition is selectively effective in *KRAS*-mutant LUAD and PDAC but not *KRAS*-WT LUAD or *KRAS*-mutant CRC. We described and discussed these results on page 10, 11, and 17 of the revised manuscript.

Together, these new results provide additional evidence supporting our finding that co-targeting FGFR and PLK1 is an effective strategy for *KRAS*-mutant lung and pancreatic cancer, but not *KRAS*-mutant colon or *KRAS*-wild type LUAD.

Please also refer to our answers to Q1 and Q4 of Reviewer #2, and Q1 of Reviewer #3.

3) Table S1 would be more informative with another column reporting drugs IC₅₀ in BEAS-2B cells as well.

We have updated Table EV1 by including IC₅₀ values of the used drugs in BEAS-2B cells.

4) Basal levels of FGFR1 detected by western blot are very faint (Fig. S3A) and make the interpretation of siRNA knock-down difficult. mRNA levels should be presented in parallel.

We measured mRNA levels by quantitative RT-PCR, which confirmed siRNA-mediated knockdown of *FGFR1* in H358 cells (**Figure EV2G**).

5) In colon cancer cell lines, PLK1 knock-down alone already exerted a very powerful therapeutic effect. It would be useful to have a table with IC50 (single agent) for all the cell lines tested in this study.

We thank the reviewer for the comment and have provided the IC50 of PLK1 inhibitors for all the cell lines tested in this study (**Table EV2**).

6) PDX number should be increased up to n=3 for sake of statistical significance (especially in light of the extreme heterogeneity of KRAS-mutant LUAD).

Thank the reviewer for the highly constructive suggestion.

We performed new experiments in two additional PDXs (PF563, PF139) of *KRAS*-mutant LUAD. Results from the PDXs confirmed that the triple modality significantly outperformed AZD4547/BI2536 combination and single agents, resulting in potent anti-tumor effects (**Figure 6F-K**).

We described the results of **Figure 6F-K** in page 14 of the revised manuscript.

7) "We noticed that genetic depletion of FGFR1 increased the expression of PLK1 and vice versa (Figure S3A)". Based on the panels presented here, this seems to be an overstatement. Proper western blot quantification should be presented in parallel.

We quantified Western blot results of PLK1, p-PLK1 (active form of the protein), and activity of PLK1 signaling (ratio of p-PLK1/PLK1/ β -Actin), which indicated that *FGFR1* knockdown markedly increased PLK1 activity in *KRAS*-mutant lung (H358, A549, H441), pancreatic (MIAPaCa-2), but not in colon (SW620) cancer cells (**Figure EV2F**). Further, siRNA-mediated PLK1 depletion upregulated FGFR1 protein in A549, H441, MIAPaca-2 and SW620 cells (**Figure EV2F**). It should be noted that H358 cells express very low levels of FGFR1 protein, so the Western blot result cannot be quantified.

Based on these observations, we changed the sentence to “we noticed that genetic depletion of FGFR1 increased PLK1 activity and PLK1 reduction upregulated FGFR1 in *KRAS*-mutant lung and pancreatic cancer cells”, on page 9 of the revised manuscript.

8) TCGA analysis on a cohort of *KRAS*-mutant LUAD (n=141) revealed a negative correlation between PLK1 mRNA and FGFs mRNA. Does this stand true also in PDAC? What about EGFR or BRAF-mutant LUAD? Or *KRAS*-mutant CRC?

We extended the analysis to PDAC, *EGFR* or *BRAF*-mutant LUAD, and *KRAS*-mutant CRC. The results were shown in **Figure EV3B-E** and described on page 9 of the revised manuscript.

9) mTORC suppression upon treatment with CH5183284 should be confirmed by siRNA against FGFR1 (Fig. S3).

We performed siRNA-based FGFR1 knockdown in A549 cells (**Figure EV4C**). Western blot confirmed that FGFR1 knockdown suppressed mTORC1, indicated by decrease of p-mTOR, p-S6 in *FGFR1*-depleted A549 cells compared with those in the control cells (**Figure EV4C**).

We described the results on page 10 of the revised manuscript.

10) Western blot analysis of PDX samples (Fig. S4D) is poor and quality should be improved. Also, full proteins panel (as in Fig. 4A) should be presented.

We extended Western blot analysis of PDX samples, with the improved results shown in **Figure EV4H**. Notably, the combination treatment not only inhibited FGFR1 signaling (reduced p-FRS2), but also substantially increased p-JNK, γ H2AX, p-E2F1, and

concomitantly decreased anti-apoptotic BCL-2 compared to single agents (**Figure EV4H**). These results confirm our in vitro observations and provide in vivo evidence supporting our conclusion.

We described the results on page 11 of the revised manuscript.

11) Are ROS levels affected by treatment with SP600125 or SB203580? And by E2F1 knock-down?

Our new experiments showed that treatment with SP600125 or SB203580, alone or in combination, did not affect ROS in H358 cells (**Figure EV4P**), as did E2F1 knockdown (**Figure EV4Q**). These observations are consistent with the notion that JNK/p38 and E2F1 act downstream of ROS.

We described these results on page 12 of the revised manuscript.

12) In vivo data presented in Fig. 6 are impressive. It would be very informative to have residual tumors analyzed for autophagy markers, ROS levels, p38 pathway, E2F1 levels.

We thank the reviewer for the constructive suggestion.

Western blot showed that AZD4547/BI6727 combination and triple AZD4547/BI6727/HCQ markedly increased p-JNK, and, to a lesser extent, p-E2F1 in residual tumors compared with single agents (**Figure EV5D**). The AZD4547/BI6727 combination also induced autophagy in residual tumors, consistent with our in vitro results (**Figure EV5D**). Interestingly, the triple combination further elevated LC3-II compared with AZD4547/BI6727 (**Figure EV5D**), indicating decreased autophagosome-lysosome fusion after HCQ treatment (Mauthe et al, 2018).

Because we were unable to identify a ROS marker used for Western blot, we instead measured ROS in murine KP cells (*Kras*^{G12D};*Trp53*^{-/-}) derived from the GEM model of *KRAS*-mutant LUAD (DuPage et al, 2009) used in our study (**Figure 7A-F**). Indeed, AZD4547/BI6727 combination remarkably upregulated ROS in KP cells, and the presence of HCQ further increased AZD4547/BI6727-induced ROS levels in the cells (**Figure EV5E**), reinforcing our *in vitro* results.

We described the above data on page 14 of the revised manuscript.

DuPage M, Dooley AL, Jacks T. Conditional mouse lung cancer models using adenoviral or lentiviral delivery of Cre recombinase. *Nat Protoc.* 2009;4:1064–72.

Mauthe M, Orhon I, Rocchi C, Zhou X, Luhr M, Hijlkema KJ, Coppes RP, Engedal N, Mari M, Reggiori F et al (2018) Chloroquine inhibits autophagic flux by decreasing autophagosome-lysosome fusion. *Autophagy* 14:1435-1455

13) Panel S3E should be cited in the text before S4A.

In the revised manuscript, we cited Figure S3E (new **Figure EV1F**) in the text before Figure S4A (new **Figure EV4D**).

Referee #2 (Remarks for Author):

In the current manuscript, the authors identify a novel synergistic combination of FGFR1 and PLK1 inhibitors which was effective KRAS mutant NSCLC and PDAC but not CRC. They demonstrated that the combination was effective in vitro and in vivo. The combination lead to increases apoptosis and ROS production as well activation/induction of JNK/p38 and E2F1. Furthermore, they demonstrated that autophagy plays a protective role in the presence of FGFR1 and PLK1 inhibition and the combination of FGFR1/PLK1 with autophagy inhibition was the most efficacious in vitro and in vivo. In summary, this is a well-written, logical and well-performed study that has identified a novel synergistic combination that is effective against KRAS mutant NSCLC and PDAC. This manuscript could be improved if the mostly minor concerns below are addressed.

We thank the reviewer for the positive assessment of our study.

1. The authors demonstrate that multiple processes that may be necessary for the combination to inhibit growth including ROS, apoptosis and G2/M. It would be helpful to define the requirement of apoptosis though testing the combination in the presence or absence of a caspase inhibitor. Furthermore, it would be helpful to see induction of ROS, JNK/p38, E2F1, G2/M arrest and apoptosis occur in the KRAS mutant CRC lines for which the combination does not cause growth inhibition.

We thank the reviewer for the insightful comments and suggestions.

First, we performed new experiments to test how the pan-apoptosis inhibitor (Q-VD-OPh) affects the efficacy of combined FGFR1/PLK1 inhibition. In *KRAS*-mutant lung cancer cells (H358), the presence of Q-VD-OPh compromised the anti-proliferative effect of the combination, evidenced by decrease of CI-PARP (**Figure 3F**), reduced percentage of

apoptotic cells (**Figure 3G**) and improved survival (**Figure 3H**), indicating that apoptosis is indeed required for the inhibitory effect of the combination.

Secondly, our new results showed that *KRAS*-mutant colon cancer cells (SW620) display a greater dependency on PLK1 for survival than *KRAS*-mutant lung and pancreatic tumor cells, as single treatment with PLK1 inhibitors (e.g., BI2536) was sufficient to induce high levels of ROS, activate the p38/JNK/E2F1 axis and promote PARP cleavage (CI-PARP) (**Figure EV4F, 4I**). These observations explain why the synergy of FGFR and PLK1 inhibitors observed in *KRAS*-mutant lung and pancreatic cancer cells does not translate to *KRAS*-mutant colon cancer cells.

Please also refer to our answers to Q4 of the same reviewer, Q2 of Reviewer #1, and Q1 of Reviewer #3.

2. The experiments in Fig 4B-F should be verified in a 2nd *KRAS* mutant cell line.

We performed new experiments in *KRAS*-mutant lung cancer cell line (A549) and the results were shown in the updated **Figure 4B-F**.

Specifically, addition of SP600125 and SB203580, inhibitors of JNK and p38, respectively, dramatically compromised the efficacy of AZD4547/BI2536 combination in A549 cells, evidenced by marked decrease of p-p38, p-JNK, p-E2F1 and CI-PARP (**Figure 4B**) and increased cell viability (**Figure 4C**).

RNAi-based *E2F1* knockdown in A549 cells showed similar results, indicated by reduced expression of p-E2F1 and CI-PARP (**Figure 4D**), increased viability (**Figure 4E**), and decreased apoptosis (**Figure 4F**) in *E2F1*-depleted A549 cells compared to control cells after treated with AZD4547/BI2536 combination.

These results are in line with those from H358 cells, further supporting the notion that activation of JNK/p38 and E2F1 contributes to the efficacy of AZD4547/BI2536 combination.

We described these results on page 12 of the revised manuscript.

3. The authors state that NAC decreased cl-PARP in H358 and A549 cell lines but the data presented in Fig. 3E is unconvincing for A549.

The improved Western blot results were presented, which showed that NAC indeed decreased CI-PARP in A549 cells (**Figure 3E**).

4. A short discussion in the DISCUSSION section of why KRAS mutant CRC is not responsive to the combination would be helpful.

In “Discussion”, we briefly discussed why *KRAS*-mutant CRC is not responsive to the combination. Please see page 17 of the revised manuscript.

Please also refer to our answers to Q1 of the same reviewer, Q2 of Reviewer #1, and Q1 of the Reviewer #3

5. The authors should describe the histology and cancer subtype of the PDX used in this study both in the results section and methods.

We have provided the histology and cancer subtype of the PDXs used in this study. See page 21 in “Material and Methods”.

Referee #3 (Remarks for Author):

In this study, the authors investigated drug combinations that could lead to more efficacious treatment for KRAS mutant lung cancer cells. Building on previous synthetic lethal studies identifying KRAS mutant cells are more sensitive to PLK1 inhibitors, the authors carried out a small drug combination screen using 21 compounds and identified the FGFR inhibitor AZD4547 to exhibit combination synergy with the PLK1 inhibitor BI2536 in KRAS mutant lung cancer cell lines in vitro and in KRAS mutant lung tumor models in vivo. Mechanistically, combined FGFR and PLK1 inhibition leads to elevated ROS production and autophagy, and inhibition of autophagy using HCQ could further enhance cell killing in a triple-drug combination. Single agent PLK1 inhibitors have met with limited success in clinical trials thus far, and this study presented pre-clinical evidence for a new combination therapy with translational potential.

We thank the reviewer for the cohesive summary and positive assessment of our study.

Major points

1. For Figure 2, the combination synergy experiment in the cell line panel was carried out using a single concentration of AZD4547 (5uM) and BI2536 (5nM). This raises the question of whether at the concentration used these inhibitors are effectively blocking their targets in different cell lines, since presumably the IC50s of these drugs are different amongst different cell lines. It is thus possible that, for KRAS mutant colorectal cell lines and KRAS WT cell lines, the lack of synergy could be due to lack of target inhibition at the chosen drug concentrations. The authors should present evidence that AZD4547 is inhibiting FGFR signaling (for examples, by pFGFR and pFRS2 blots), and BI2536 is inhibiting PLK1 activity (for example by using pPLK1 blot) in representative cell lines from this panel to rule out this possibility.

We thank the reviewer for the very insightful comments. In the revised manuscript, we provided different lines of evidence that supports our observations in **Figure 2**.

First, viability assay that assesses dose-response curves across a wide range of AZD4547 (up to 20 μ M) and BI2536 (up to 20 nM) confirmed a strong synergy between AZD4547 and BI2536 in *KRAS*-mutant lung (H358, H441, A549) and pancreatic (MIAPaCa-2), but not in SW620, DLD-1, nor in EBC-1 or H1993 cells (**Figure EV2C, D**).

Secondly, Western blot showed that AZD4547 (5 μ M) inhibited FGFR signaling (decrease in p-FRS2, p-AKT) in SW620 and EBC-1 cells, as did BI2536 (5 nM) that reduced p-PLK1 (**Figure EV4I**). However, the combination showed similar effects on the expression of p-p38, p-JNK and p-E2F1 as single agents (AZD4547, BI2536) alone in SW620 and EBC-1 cells (**Figure EV4I**). Consequently, the combination treatment failed to increase PARP cleavage compared to BI2536 alone in SW620 cells, or did not induce PARP cleavage at all in EBC-1 cells (**Figure EV4I**).

Thirdly, the combination did not significantly increase ROS levels in SW620 cells compared to single agents alone; in EBC-1 cells, the ROS level induced by the combination was only slightly higher than AZD4547 or BI2536 (**Figure EV4F**).

Thus, the results from SW620 and EBC-1 cells show a different scenario from those observed in *KRAS*-mutant lung and pancreatic cancer cells (**Figure 3B, 3E, 4A**), which may provide a mechanistic explanation why the synergy between FGFR1/PLK1 inhibitors does not occur in *KRAS*-mutant colon or *KRAS*-WT LUAD.

Please also refer to our answers to Q2 of Reviewer #1, and Q1 and Q4 of Reviewer #2.

2. Could NAC rescue the toxicity of the triple combination of AZD, BI and HCQ?

We performed new experiments, which showed that addition of NAC substantially rescued the toxicity of the triple combination in H358 and A549 cells (**Figure EV4J**).

Minor points

1. Does FGFR1/2/3 or FRS2 expression differ between KRAS mutant and KRAS WT cell lines?

Western blot showed that FGFR1/2/3 and FRS2 expression differ in *KRAS*-mutant (A549, H358, MIAPaCa-2, and SW620) and *KRAS*-WT (EBC-1, H1993) cancer cells. Specifically, A549 and SW620 expressed FGFR1, MIAPaCa-2 expressed FGFR1/2/3, and A549, H358, MIAPaCa-2, and SW620 expressed FRS2 (**Figure EV2J**).

2. Figure 2E. A WB confirming FGFR1 and PLK1 siRNA knockdown should be shown.

Our WB results (**Figure EV2F**) confirmed siRNA-mediated knockdown of FGFR1 and PLK1 in A549, H441, MIAPaCa-2 and SW620 cells. For H358 cells expressing low levels of FGFR1 (**Figure EV2F,J**), *FGFR1* mRNA was determined by qRT-PCR (**Figure EV2G**).

3. Figure 2F. A WB confirming FGFR1 signaling inhibition in PDX BE564T should be shown.

WB analysis of residual tumors (PDX BE564T) was shown in **Figure EV4H**, which indicated that AZD4547 alone and the combination (AZD4547 plus BI2536) effectively inhibited FGFR1 signaling, indicated by marked decrease in p-FRS2 (**Figure EV4H**).

4. Figure 3A. for the GSEA analysis, Why was a different FGFR inhibitor (CH5183284) used and why were H1581 and H520 cells used? These two cell lines did not appear in the cell line panel study in Figure 2. Does the FGFRi+PLK1i combination exhibit drug synergy in these two cell lines?

In **Figure 3A**, we performed GSEA based on a previously published study (Nakanishi et al, 2015), whereby the transcriptomes of *FGFR1*-amplified H520 and H1581 cells treated with the FGFR inhibitor CH5183284 were analyzed.

Given that KRAS is a downstream effector of FGFR1 signaling, we assume that oncogenic activation of KRAS or FGFR1 (e.g., *FGFR1* amplification) might deregulate similar cellular processes. Indeed, we have recently showed that FGFRi/PLK1 combination exhibited strong synergy in H520 and H1581 cells (Zhang et al, 2021).

Nakanishi Y, Mizuno H, Sase H, Fujii T, Sakata K, Akiyama N, Aoki Y, Aoki M, Ishii N. ERK Signal Suppression and Sensitivity to CH5183284/Debio 1347, a Selective FGFR Inhibitor. *Mol Cancer Ther* 2015; 14: 2831-2839

Yang Z, Liang SQ, Yang H, Xu D, Bruggmann R, Gao Y, Deng H, Berezowska S, Hall SRR, Marti TM, Kocher GJ, Zhou Q, Schmid RA, Peng RW. CRISPR-mediated kinome editing prioritizes a synergistic combination therapy for FGFR1-amplified lung cancer. *Cancer Res*. 2021 Mar 8;canres.2276.2020. doi: 10.1158/0008-5472.CAN-20-2276.

5. Figure 5F. ATG5 siRNA knockdown efficiency appears to be poor.

The improved Western blot result of ATG5 knockdown was shown in **Figure 5F**.

6. Figure 6E. Do the double and triple drug combinations result in a difference in tumor grade in the KP model compared to single agents or control?

We analyzed tumor grade in KP mouse model according the protocol described by DuPage et al. (2009). This analysis revealed that the triple drug combination indeed resulted in a difference in tumor burden, indicated by marked decrease of “Grade 3-4” and increase of “Grade 1” compared to single agents and control (**Figure 7E**). This observation is consistent with the in vitro and in vivo results, providing additional evidence supporting our conclusion.

DuPage M, Dooley AL, Jacks T. Conditional mouse lung cancer models using adenoviral or lentiviral delivery of Cre recombinase. *Nat Protoc*. 2009; 4(7):1064-72.

Dear Prof. Peng,

Thank you for the submission of your revised manuscript to EMBO Molecular Medicine. I am pleased to inform you that we will be able to accept your manuscript pending the following final amendments:

1) Figures: Please add scale bars to Figure 6D and EV 2K. Scale bars in Figure 5 are not readable, please correct and add readable scale bars.

2) Supplementary Figures and Tables: EV Figures should either be uploaded as separate files with their legend in the main manuscript or keep them in one file as they are now, rename the file to Appendix and add table of content on the first page. Please place all EV Tables to the same Appendix file. Nomenclature in the manuscript text needs to be updated from Figure EV1 to Appendix Figure S1 etc. and from Table EV1 to Appendix Table S1 etc. Please check "Author Guidelines" for more information.

<https://www.embopress.org/page/journal/17574684/authorguide#expandedview>

3) In the main manuscript file, please do the following:

- Correct/answer the track changes suggested by our data editors by working from the attached/uploaded document.
- Make sure that all special characters display well.
- In M&M, provide the antibody dilutions that were used for each antibody.
- In M&M, include a statement that informed consent was obtained from all human subjects and that the experiments conformed to the principles set out in the WMA Declaration of Helsinki and the Department of Health and Human Services Belmont Report.
- In M&M, the statistical paragraph should reflect all information that you have filled in the Authors Checklist, especially regarding randomization, blinding, replication.
- Please remove "Funding" information from the title page and list all sources of funding only in "Acknowledgements".

4) Synopsis: Every published paper now includes a 'Synopsis' to further enhance discoverability. Synopses are displayed on the journal webpage and are freely accessible to all readers. They include separate synopsis image and synopsis text.

- Synopsis image: Please provide a striking image or visual abstract as a high-resolution jpeg file 550 px-wide x (250-400)-px high to illustrate your article.
- Synopsis text: Please provide a short stand first (maximum of 300 characters, including space) as well as 2-5 one sentence bullet points that summarise the paper as a .doc file. Please write the bullet points to summarise the key NEW findings. They should be designed to be complementary to the abstract - i.e. not repeat the same text. We encourage inclusion of key acronyms and quantitative information (maximum of 30 words / bullet point). Please use the passive voice.

5) Source data: We encourage you to include the source data for figure panels that show essential data. Numerical data should be provided as individual .xls or .csv files (including a tab describing the data). For blots or microscopy, uncropped images should be submitted (using a zip archive if multiple images need to be supplied for one panel). Please check "Author Guidelines" for more information. <https://www.embopress.org/page/journal/17574684/authorguide#sourcedata>

6) As part of the EMBO Publications transparent editorial process initiative (see our Editorial at <http://embomolmed.embopress.org/content/2/9/329>), EMBO Molecular Medicine will publish online a Review Process File (RPF) to accompany accepted manuscripts. This file will be published in conjunction with your paper and will include the anonymous referee reports, your point-by-point

response and all pertinent correspondence relating to the manuscript. Let us know whether you agree with the publication of the RPF and as here, if you want to remove or not any figures from it prior to publication. Please note that the Authors checklist will be published at the end of the RPF. 7) Please provide a point-by-point letter INCLUDING my comments as well as the reviewer's reports and your detailed responses (as Word file).

I look forward to reading a new revised version of your manuscript as soon as possible.

Yours sincerely,

Zeljko Durdevic

***** Reviewer's comments *****

Referee #2 (Remarks for Author):

The authors have sufficiently addressed my concerns. Nice Work!

The authors performed the requested editorial changes.

We are pleased to inform you that your manuscript is accepted for publication and is now being sent to our publisher to be included in the next available issue of EMBO Molecular Medicine.

Corresponding Author Name: Ren-Wang Peng, Ralph A. Schmid
Journal Submitted to: EMBO Molecular Medicine
Manuscript Number: EMM-2020-13193